



# The effect of hydrology and crevasse wall contact on calving

Maryam Zarrinderakht[1], Christian Schoof[1], and Anthony Peirce[2]

[1]Department of Earth, Ocean and Atmospheric Sciences, University of British Columbia, BC, Canada
[2]Department of Mathematics, University of British Columbia, BC, Canada

**Correspondence:** M. Zarrinderakht (mzaryam@eoas.ubc.ca)

**Abstract.** Calving is one of the main controls on the dynamics of marine ice sheets. We solve a quasi-static linear elastic fracture dynamics problem, forced by a viscous pre-stress describing the stress state in the ice prior to the introduction of a crack, to determine conditions under which an ice shelf can calve for a variety of different surface hydrologies. Extending previous work, we develop a boundary-element-based method for solving the problem, which enables us to ensure that the faces of crevasses are not spuriously allowed to penetrate into each other in the model. We find that a fixed water table below the ice surface can lead to two distinct styles of calving, one of which involves the abrupt unstable growth of a crack across a finite thickness of unbroken ice that is potentially history-dependent, while the other involves the continuous growth of the crack until the full ice thickness is cracked, which occurs at a critical combination of extensional stress, water level and ice thickness. We give a relatively simple analytical calving law for the latter case. For a fixed water volume injected into a surface crack, we find that complete crack propagation almost invariably happens at realistic extensional stresses if the initial crack length exceeds a shallow threshold, but we also argue that this process is more likely to correspond to the formation of a localized, moulin-like slot that permits drainage, rather than a calving event. We also revisit the formation of basal cracks, and find that, in the model, they invariably propagate across the full ice shelf at stresses that are readily generated near an ice shelf front. This indicates that a more sophisticated coupling of the present model (which has been used in a very similar form by several previous authors) needs modification to incorporate the effect of torques generated by buoyantly-modulated shelf flexure in the far field.

## 1   Introduction

Calving is the formation of fractures that separate newly formed icebergs or smaller pieces of ice from a contiguous ice shelf or marine-terminating glacier. In marine ice sheet and outlet glacier models, the choice of a 'calving law' has a significant effect on steady state configurations (Schoof et al., 2017; Haseloff and Sergienko, 2018). Despite its importance, there is currently no comprehensive theory for calving.

A variety of different approaches have been used to model fracture crevasse, in ice. Aside from early heuristic "zero-stress" type models (Nye, 1957; Nick et al., 2010), these are primarily discrete element models (which do not pretend to represent ice as a continuum), linear elastic fracture mechanics models, which focus on one or a few discrete cracks, and continuum damage mechanics models, which treat calving as the result of the density of microfractures accumulating to generate a macroscopic crevasse that penetrates through the ice thickness (Larour and Aubry, 2004; Benn et al., 2007; Borstad et al., 2012, 2013; Krug



et al., 2014; Mobasher et al., 2016; Yu et al., 2017; Benn et al., 2017; Todd et al., 2018). In addition to different assumptions about the basic physics involved, there are additionally different numerical approaches that can be applied to the resulting models, especially in case of linear elastic fracture mechanics (Tsai and Rice, 2012; Lipovsky, 2020).

One of the most significant challenges is to capture fracture evolution in an ice sheet or ice shelf that flows viscously over long time scales (Yu et al., 2017), owing to the very different time scales and physical processes involved. Damage mechanics attempts to bridge that gap with a smoothly evolving damage function describing fracture density. By contrast, direct application of linear elastic fracture mechanics only attempts to capture short-term changes in stress during the formation of a fracture, and their role in fracture propagation. In this paper, we take the latter approach, studying how elastic fracture

propagation over short time scales is controlled by a viscous pre-stress, overburden, and water pressure in a crevasse.

Our model is a generalization of the crack penetration models for ice shelves in van der Veen (1998a,b) and Lai et al. (2020). Like these authors, we focus on vertical fracture propagation in two dimensions under plane strain conditions, omitting complications introduced by three-dimensional rift formation (Lipovsky, 2020), and consider cracks that are far from either the ice front or the grounding line. We formulate the model (and the corresponding numerical solution method) for arbitrary

two-dimensional geometries, and impose contact constraints that prevent opposite sides of a crack from interpenetrating. In particular, we extend the completely fluid-filled and completely dry crack scenarios in Lai et al. (2020) to more general hydrologies, focusing on surface cracks in which the water level is either prescribed, or constrained by a finite volume injected into the crack. In addition, we revisit the case of water-filled basal cracks previously studied in van der Veen (1998b) and Lai et al. (2020).

In order to deal with crack face contacts and, ultimately, with general ice geometries, we have to abandon the use of tabulated Green's functions previously pioneered in van der Veen (1998a,b) and Lai et al. (2020). In this paper, we employ a boundary element method to compute stress fields in the ice. We show how possible steady state crack configurations change when stress, thickness, and hydrological forcing parameters are varied, and use these results to derive calving laws for two-dimensional ice shelves. Depending on parameter combinations, we find that calving can either occur by steady state crack lengths continuously

growing to span the entire ice thickness, or by a steady state crack that partially penetrates the ice being destabilized and growing across the full thickness of the ice.

The paper is organized as follows: We formulate the fracture model of van der Veen (1998a,b) and Lai et al. (2020) in terms of partial differential equations in section 2.1, separating the elastic stress field induced by the introduction of a fracture from a viscous pre-stess, and accounting for contact constraints that prevent crack faces from interpenetration. We also formulate

the crack propagation criterion we use in section 2.2. Following Lai et al. (2020), we reduce the parameter space of the model through non-dimensionalization in section 2.3 and sketch the numerical method used in section 3, with further detail relegated to appendix A. Results are presented in section 4, where we focus on the case of a single crack incised into a parallel-sided slab of ice; in that case, the visocus pre-stress is known exactly, and more general ice geometries will be considered in a separate paper. As in van der Veen (1998a,b) and Lai et al. (2020), we begin by studying the dependence of the stress intensity

factor on crack length in section 4.1, identifying the range of steady states accessible for individual parameter combinations. We consider the effect of incorporating the contact constraints into the model in sections 4.2–4.3. Section 4.4 systematically





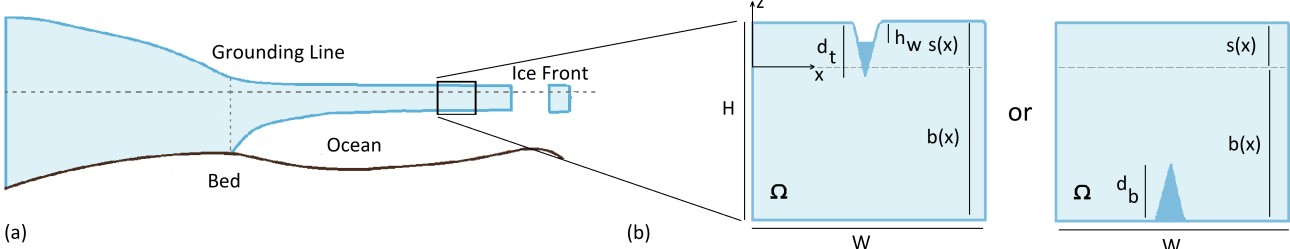

**Figure 1.** Cross section geometry of a marine ice sheet, and the geometry of the problem: part of a floating ice shelf with a surface or bottom crevasse.

explores the dependence of steady state configurations on parameter variations, allowing us to describe how changes in forcing can precipitate calving in section 4.5. In section 5, we use these results to formulate calving laws that can be used in large scale models, focusing on the distinction between calving laws that require a knowledge of the history of the ice shelf from those that can be formulated purely in terms of current forcing parameters in the form of extensional stress, ice thickness, and hydrology. We summarize our findings and point to some of their limitations in section 6, where we identify torque-driven calving as being poorly represented in the model as formulated, which does not incorporate the effect of changes buoyancy due to vertical deflections.

## 2 Model

### 2.1 Model Description

We employ a Cartesian coordinate system $(x,z) = (x_1, x_2)$, with a horizontal $x$-axis and $z$ measured relative to sea level. We denote the domain by $\Omega$ and its boundary by $\partial\Omega$, with outward-pointing unit normal $\mathbf{n}$. In order to cast the problem previously solved by van der Veen (1998a,b) and Lai et al. (2020) the form of partial differential equations (appropriately generalized to take greater account of variations in surface hydrology, and allowing for contact between crack faces as described below), it is necessary to assume a compressible Maxwell-type viscoelastic rheology, and separate the stress tensor $\sigma_{ij}$ into a viscous pre-stress $\sigma_{ij}^v$ that existed just prior to crack propagation, and an elastic stress $\sigma_{ij}^e$ generated by crack propagation on time scales much faster than a single Maxwell time (Christensen, 1971),

$$\sigma_{ij} = \sigma_{ij}^v + \sigma_{ij}^e. \tag{1}$$

In common with many other fracture problems (Zehnder, 2012; Crouch and Starfield, 1983), we consider only the quasi-static case here, so

$$\frac{\partial \sigma_{ij}}{\partial x_j} + \rho_i g_i = 0, \tag{2}$$

where $\mathbf{g} = (0, -g)$ is acceleration due to gravity and the summation convention is used; (2) is equally applicable to the equivalent, purely viscous flow problem that applies at longer time scales.





The the upper surface at $z = s$ (Figure 1) is traction-free, therefore

$$\sigma_{ij}n_j = 0. \tag{3}$$

At the lateral boundaries at $x_1 = 0, W$, we impose a normal stress, the sum of a cryostatic contribution and an imposed extensional (or 'resistive') stress $R_{xx}$ (equal to $4\mu\partial U/\partial x$ if $\mu$ is viscosity and $U$ is far field ice velocity, see also (van der Veen, 1998a,b))

$$\sigma_{1j}n_j = (R_{xx} - \rho_i g(s - z))n_i. \tag{4}$$

At the base of the ice $z = b$, we have hydrostatic pressure from the ocean

$$\sigma_{ij}n_j = \rho_w g b n_i, \tag{5}$$

Note that, in this paper, we will solve the model only for the case of cracks incised into a wide slab of ice whose upper and lower surfaces before crack formation are parallel, as is also the case in van der Veen (1998a,b) and Lai et al. (2020); the numerical method described is however equally suited to more general geometries, which we will consider in a separate paper.

For the parallel-sided slab, the lateral stress field field in (4) naturally results from the viscous deformation of a wide slab of ice in which ice flows as a plug with no vertical shear, for which the viscous stress tensor is simply

$$\sigma_{11}^v = R_{xx} - \rho_i g(s - z), \qquad \sigma_{12}^v = \sigma_{21}^v = 0, \qquad \sigma_{22}^v = -\rho_i g(s - z). \tag{6}$$

Importantly, the stress field defined by (6), which on its own satisfies lateral stress conditions (4), cannot be generated by an elastic rheology (in the sense that there is no displacement field that generates $\sigma_{ij}^v$ through (7)–(8) below), which explains our

our insistence on separation of $\sigma_{ij}$ into viscous and elastic parts $\sigma_{ij}^v$ and $\sigma_{ij}^e$, respectively.

For the domain used in section 4 in this paper, (6) is therefore the appropriate form of the visocus pre-stress; for a more general initial geometry, it is necessary to solve (2)– (5) on the uncracked domain first, subject to a purely viscous rheology (that is, putting $\sigma_{ij} = \sigma_{ij}^v$) in order to find the viscous pre-stress before introducing the cracks. We will deal with that more complicated procedure in a separate paper (see also Yu et al. (2017)).

Assuming the viscous pre-stress is known and satisfies the boundary equations (3)–(5), our focus will be on finding the elastic stress field that results from the introduction of cracks. Our model only accounts for an in-plane displacement field $\mathbf{u}(x, z) = (u_x, u_z) = (u_1, u_2)$ with an associated strain

$$\varepsilon_{ij} = \frac{1}{2}\left(\frac{\partial u_i}{\partial x_j} + \frac{\partial u_j}{\partial x_i}\right). \tag{7}$$

We assume that $\sigma_{ij}^e$ is described by an isotropic, linear elastic rheology. Assuming small strains, this relates stress $\sigma_{ij}$ to strain

$\varepsilon_{ij}$ in plane strain conditions as

$$\varepsilon_{ij} = \frac{1}{E'}\left(\frac{1}{1-\nu}\sigma_{ij}^e - \frac{\nu\sigma_{kk}^e\delta_{ij}}{1-\nu}\right), \tag{8}$$



the sum over repeated indices running over $\{1,2\}$. Here, $\nu$ is Poisson's ratio and $E' = E/(1-\nu^2)$ is the plane strain modulus, $E$ being Young's modulus, with both $\nu$ and $E'$ assumed to be constant.

Cracks are internal boundaries on which displacement may be discontinuous, and stress boundary conditions can be pre-
scribed. We define an outward-pointing unit normal $\mathbf{n}^\pm$ to each side of the crack, denoting the left by the superscript $+$ and the right by $-$. By outward-pointing, we mean that, on each side, $\mathbf{n}^\pm$ points towards the crack rather than the interior of the domain. In the small-strain limit of linear elasticity, the two sides of the crack are parallel and $\mathbf{n}^+ = -\mathbf{n}^-$. Similarly denot-
ing $\mathbf{v}^\pm$ as the relevant limit of an arbitrary vector field $\mathbf{v}$, we can define a jump in its normal component across the crack as $[v]_-^+ = \mathbf{v}^+ \cdot \mathbf{n}^+ + \mathbf{v}^- \cdot \mathbf{n}^-$, or equally, in subscript notation $[v_i]_-^+ = v_i^+ n_i^+ + v_i^- n_i^-$. In that notation, crack width $w$ is

$$w = -[\mathbf{u}]_-^+. \tag{9}$$

Boundary conditions on either the top or bottom crack can be expressed as

$$\text{either} \qquad (w > 0 \text{ and } -\sigma_{ij} n_i n_j = p_f) \qquad \text{or} \qquad (w = 0, \quad [\sigma_{ij} n_j]_-^+ = 0, \text{ and } -\sigma_{ij} n_i n_j \geq p_f) \tag{10}$$

where $p_f$ is fluid pressure in the crack. These conditions ensure that normal stress is continuous and exceeds fluid pressure where the crack is closed, or equals fluid pressure where the crack is open. In addition, we assume that shear stress vanishes even when the crack walls re-contact, thus assuming them to be smooth and not subject to healing on the time scale under consideration:

$$(\delta_{ij} - n_i n_j)\sigma_{ij} n_j = 0, \tag{11}$$

The stress conditions above hold for both faces of the crack, in the sense of $\sigma_{ij}$ being evaluated as the limit taken from either side of the crack, with $\mathbf{n}$ being the outward-pointing unit normal that corresponds to the side from which the limit is taken. Here $p_f$ is the fluid pressure inside any of the cracks, given by

$$p_f = \max(\rho_w g(s - h_w - z), 0), \tag{12}$$

in the top crack, and

$$p_f = -\rho_w g z \text{ if } z \leq 0, \qquad p_f = 0 \text{ if } z > 0 \tag{13}$$

in the bottom crack. $h_w$ is the depth of the water level in the surface crevasses below the upper surface $z = s$ in the surface crevasses.

We consider two basic scenarios. The first involves a prescribed water level $h_w$ below the ice surface as previously used by van der Veen (1998a), while the second involves a prescribed water volume in the top crack. The latter is motivated by Nick et al. (2010), who use the somewhat more difficult-to-justify assumption of a prescribed water column height above the bottom of the crevasse (see also Schoof et al., 2017) one would not generally expect the column height to be prescribed in nature, while water volume might be. As we will see in this paper, we obtain very different behaviour depending on whether water level or water volume is prescribed.





Let $V_w$ be the prescribed water volume and $d_t$ the vertical extent of the crack (see Figure 1). If all the prescribed water volume can be accommodated in the crack, then there exists an $h_w > 0$ such that

$$V_w = \int_{s-d_t}^{s-h_w} w(z)\mathrm{d}z. \tag{14}$$

Otherwise, if the proscribed volume cannot be accommodated in the crack, then $h_w = 0$ and the excess is stored at the surface. Note an important caveat to (12)–(13): both assume that negligible hydraulic potential gradients are required to drive fluid flow along the cracks as their tips move and the cracks open or close, so that water pressure can be treated as hydrostatic in each crack. The propagation criteria in the next section are built around this assumption, which gives a particularly simple way of handling the stability of cracks but likely needs to be superseded by a more sophisticated treatment of water movement in the 150 cracks in future work.

We will refer to the constraint requiring non-negative crack width, $w \geq 0$, as the "contact constraint" in the remainder of this paper. Note that van der Veen (1998a,b) and Lai et al. (2020) do not enforce the contact constraint. To reproduce their results, we also consider an alternative (and not always physically viable) set of boundary conditions on the crack, putting

$$\sigma_{ij}n_j = -p_f n_i. \tag{15}$$

instead of (10); the prescription of a fixed water volume or water level remains as described above in that case.

Before we move onto crack propagation in the next subsection, note also the following: in common with van der Veen (1998a,b) and Lai et al. (2020), our formulation does not consider the effect of elastic displacements on the position of the upper or lower boundaries, and therefore on changes in water pressure and hence in buoyant support. Our elastic model is based on small strains, which implies that $E^{-1}\rho_i g H \ll 1$; with ice thicknesses around $H \approx 1000$ m, $g \approx 10$ m s$^{-2}$, $\rho_i \approx 900$ kg m$^{-3}$ 160 and a Young's modulus of $E \approx 10^9$ Pa (Vaughan, 1995), we find strains around $10^{-2}$. Over the length scale of given by a single ice thickness (relevant to elastic deformation around a crack that penetrates partially through the ice), this implies that elastic displacements are small and buoyant effects are higher order corrections: a consideration of buoyant effects requires flexure over a longer horizontal length scale $(\rho_i g H/E')^{1/4} H \gg H$ (Wagner et al., 2016; Buck and Lai, 2021); formally, deformation at that scale should couple to the model described here via far-field boundary conditions at the lateral sides of the domain 165 through the procedure of asymptotic matching. As in prior work by van der Veen (1998a,b) and Lai et al. (2020), we do not consider that complication here, opting for the simple boundary conditions (4) instead. We return to the limitations this imposes in section 6.

With the simpler boundary conditions (4) lacking any far-field torque or net shear force, note also that the model above is subject to the following Archimedean flotation solvability condition, which can be derived by integrating (2) over the domain 170 and using the divergence theorem:

$$\rho_i \int_0^W (s-b)\mathrm{d}x = -\rho_w \int_0^W b\,\mathrm{d}x; \tag{16}$$





for a parallel-sided slab geometry of thickness $H = s - b$ with constant $s$ and $b$, this gives the simple and familiar $s = (1 - \rho_i/\rho_w)H$, $b = -(\rho_i/\rho_w)H$.

## 2.2 Propagation Criterion

We follow linear elastic fracture mechanics (Zehnder, 2012) in assuming that fracture propagation can be described by a simple fracture toughness. Near the tip of the crack, the stress field generally becomes singular (except when the crack faces touch with $w = 0$). The stress intensity factor for a mode I crack being computed as

$$K_I = \lim_{r \to 0} \sqrt{2\pi r} \sigma_{\theta\theta}(r, 0), \tag{17}$$

in a local $(r, \theta)$ polar coordinate system centered on the crack tip, $\theta = \pi$ being tangential to the crack. Crack propagation is
assumed to be controlled by a constant fracture toughness $K_{Ic}$, with measured values of $K_{Ic}$ for polycrystalline ice lying between 0.1 MPa.m$^{1/2}$ to 0.4 MPa.m$^{1/2}$ (Rist et al., 1996). The crack will not propagate if $K_I < K_{Ic}$, while the crack propagates once static $K_I$ exceeds $K_{Ic}$.

In a strict sense, the static force balance model can therefore only compute static crack lengths which are such as to ensure that $K_I \leq K_{Ic}$ (meaning, the crack may be on the point of moving but remains static), and the length of the crack then becomes
part of the solution, rather than being prescribed. It is insufficient to understand how cracks grow as the forcing on the system changes, in the form of changes to the excess tension $R_{xx}$ or the water volume $V_w$ (or equivalently, to water level depth below the surface $h_w$). If there is only a single crack, its dynamics under changes in parameters are likely to be simple: any such change that reduces the static $K_I$ below $K_{Ic}$ would leave the crack length unchanged, while increases in static $K_I$ above $K_{Ic}$ would cause lengthening of the crack until its tip once more attains the static value of $K_{Ic}$ (or the fracture propagates all the
way through the ice). This heuristic argument results in a simple stability criterion for steady state cracks, used in Lai et al. (2020): if a slight lengthening of a steady state crack results in an decrease in $K_I$ in the static stress model, then that crack is stable, while it is unstable or marginally stable otherwise. In the case of multiple cracks, however, not all cracks need to propagate simultaneously or at the same speed, and it is then unclear which cracks should be lengthened until they reach $K_{Ic}$ at which point cracks will have stress intensity factors below $K_{Ic}$ when a new equilibrium is reached.

Dynamic propagation of cracks typically reduces the stress intensity factor, and the rate of crack propagation is that which lowers the dynamic $K_I$ to the critical value $K_{Ic}$. There are two processes by which this reduction in stress intensity factor can occur: for sufficiently rapid crack propagation, inertial effects can be the dominant effect (Freund, 1990), or changes in fluid pressure in the crack driven by fluid flow as the crack tip advances and the crack widens can dominate (Spence and Sharpe, 1985). Here we investigate only the former, which is strictly applicable to dry cracks but furnishes a very simple propagation
rule. The reason for persisting with this process is that it makes the calving problem tractable, in which crack propagation has to be computed for a large set of combinations of forcing parameters.

In a general, the computation of $K_I$ during fracture propagation then requires a dynamic model in which inertial terms are not omitted in equation (2). Solving a time-dependent problem that captures elastic waves renders our just-stated objective of computing fracture propagation for many forcing parameters intractable. Short of solving a full dynamic crack propagation





problem, we can use the semi-analytical theory of Freund (1990), who considers the situation in which the statically computed $K_I$ only slightly exceeds $K_{Ic}$. In that case, inertial terms are only significant in a small boundary layer around the crack tip, and the stress field far from the crack tip can be determined using the static model described above. The stress intensity factor $K_I$ at the moving crack tip can then be related to the stress intensity factor computed from the static model $K_{I,stat}$ through

$$K_I = K_{I,stat}K(\dot{d}), \tag{18}$$

where $d$ is crack length, the dot on $d$ signifies an ordinary derivative with respect to time, and the 'universal function' $K$ as computed by (Freund, 1990); the key property here is that $K$ increases with $\dot{d}$. Assuming that $K_{I,stat} - K_{Ic}$ is small but positive so that the crack will propagate, we can linearize equation (18) as $K_{Ic} = K_{I,stat}\left(1 + K'(0)\dot{d} + O(\dot{d}^2)\right)$, where we have used the fact that the crack is propagating, so $K_I = K_{Ic}$. This leads to the simplified propagation equation.

$$\dot{d} = -\frac{K_{I,stat} - K_{Ic}}{K_{Ic}K'(0)} + O\left((K_{I,stat} - K_{Ic})^2\right), \tag{19}$$

where $K'(0) > 0$. Despite the fact that the derivation of this evolution equation for crack length strictly-speaking only applies to the case of the static stress intensity factor $K_{I,stat}$ that exceeds the fracture toughness $K_{Ic}$ by a small amount, we assume that equation (19) holds whenever $K_{I,stat} \geq K_{Ic}$ to facilitate rapid solution. When $K_{I,stat} < K_{Ic}$, the simplest assumption to make is that the crack tip does not evolve, so there is no healing of the crack, in which case we can generalize (19) as

$$\dot{d} = \max\left(-\frac{K_{I,stat} - K_{Ic}}{K_{Ic}K'(0)}, 0\right). \tag{20}$$

Note that if, as is implicitly assumed here, crack propagation occurs in a predefined direction (by symmetry vertically, in the examples in this paper), then (20) represents a dynamical system with as many dimensions as there are cracks: for a given set of crack lengths $d$, and therefore a given domain, the elastostatic problem of the previous section allows the stress field and therefore the $K_I$ at each crack tip to be computed uniquely. In other words, the $K_I$ are functions of the crack lengths $d$ as dynamic variables, as well as of the shape of the external boundaries encoded in $s$, $b$ and $W$, and of the remaining model parameters $\rho_i$, $\rho_w$, $g$, $R_{xx}$, $h_w$ or $V_w$, $E'$, $\nu$ and $K_{Ic}$. The structure of the problem as a dynamical system permits relatively easy analysis of calving in terms of the existence and stability of steady state solutions.

## 2.3 Scaling

In keeping with our goal of being able to sample parameter space widely, we can reduce the number of free model parameters in the model by non-dimensionalising, defining starred dimensionless variables through (see also Lai et al., 2020)

$$x_i = [x]x_i^*, \quad t = [t]t^*, \quad \sigma_{ij} = [\sigma]\sigma_{ij}^*, \quad u_i = [u]u_i^*, \quad \varepsilon_{ij} = [\varepsilon]\varepsilon_{ij}^*, \quad s = [x]s^*, \quad b = [x]b^*, \quad p_f = [\sigma]p_f^*, \tag{21}$$

choosing the length scale $[x]$ to be the mean $H$ over ice thickness $s - b$, and defining the remaining scales through

$$[\sigma] = \rho_i gH, \quad [\varepsilon] = \frac{[\sigma]}{E'}, \quad [u] = [\varepsilon][H], \quad [K_I] = \rho_i g[H]^{3/2}, \quad [t] = -\frac{K'(0)K_{Ic}}{\rho_i g[H]^{1/2}}. \tag{22}$$




These lead to six dimensionless parameters in addition to Poisson's ratio, of the form

$$\tau = \frac{R_{xx}}{\rho_i g H}, \quad \beta = \frac{V_w E'}{\rho_i g H^3}, \quad \eta = \frac{h_w}{H}, \quad \kappa = \frac{K_{Ic}}{\rho_i g H^{3/2}}, \quad r = \frac{\rho}{\rho_w}, \quad W^* = \frac{W}{H}. \tag{23}$$

For a general domain shape with arbitrary upper and lower surfaces $b$ and $s$ (subject to the solvability condition (16)), we obtain a scaled viscous pre-stress

$$\sigma_{ij}^{v*} = \frac{\sigma_{ij}^v}{\rho_i g H},$$

which is such that $\sigma_{ij}^{v*}$ depends only on $b^*$, $s^*$, $\tau$, $r$ and any viscous rheological parameters (which, for an isothermal Glen's law rheology (Cuffey and Paterson, 2010) would simply by the usual exponent $n$). For the specific, parallel-sided slab geometry that we are considering here, the viscous pre-stress (6) simply becomes

$$\sigma_{11}^{v*} = \tau - (s^* - z^*), \qquad \sigma_{12}^{v*} = \sigma_{21}^{v*} = 0, \qquad \sigma_{22}^{v*} = -(s^* - z^*). \tag{24}$$

With this simple geometry, the upper and lower surfaces also reduce to the simpler $s^* = (1 - r)$, $b^* = -r$.

Note that we only treat one of $\beta$ or $\eta$ as a prescribed parameter, depending on whether we are looking at a fixed water level or fixed water volume; the other is then implicitly defined through (34) below. We can estimate typical values of some of these parameters: for an unconfined ice shelf, the excess extensional stress is (Shumskiy and Krass, 1976; van der Veen, 1983; MacAyeal and Barcilon, 1988)

$$R_{xx} = (1 - r)\rho_i g H / 2, \tag{25}$$

in which case $\tau = (1 - r)/2 \approx 0.05$. For confined ice shelves, we typically expect smaller tensile stresses (Doake et al., 1998). $\eta$ is a water table depth and naturally lies between 0 (when the water level is at the ice surface) and 1 (crevasses are invariably dry). $\kappa$ is typically small: with $K_{Ic} = 0.1$ MPa.m$^{1/2}$ and $H = 500$ m, we obtain $\kappa \approx 10^{-3}$.

In terms of these dimensionless variables and parameters, omitting the asterisks on the dimensionless variables immediately, the model becomes the following: (7), (11) and (17) remain unchanged, while the remaining model equations (8), (2), (3), (5), (4), (10), (12), (14) and (20) are replaced by force balance in the form

$$\varepsilon_{ij} = \frac{\sigma_{ij}^e}{1 - \nu} - \frac{\nu \sigma_{kk}^e \delta_{ij}}{1 - \nu}, \tag{26}$$

$$\frac{\partial \sigma_{ij}^e}{\partial x_j} = 0, \tag{27}$$

inside the domain, with vanishing surface traction on the elastic part of the stress tensor on exterior boundaries

$$\sigma_{ij}^e n_j = 0, \qquad\qquad \text{at } x = 0, W^*, \text{ and at } z = s, b, \tag{28}$$

$$\tag{29}$$

while on the crack surfaces

$$\text{either} \quad (w > 0 \text{ and } -\sigma_{ij}^e n_i n_j = p_f + \sigma_{ij}^v n_i n_j), \quad \text{or} \quad (w = 0, \quad [\sigma_{ij}^e n_j]_-^+ = 0, \text{ and } -\sigma_{ij}^e n_i n_j \geq p_f + \sigma_{ij}^v n_i n_j). \tag{30}$$



Alternatively, when disabling the contact constraint $w \geq 0$, we simply have $\sigma_{ij}^e n_i n_j = -p_f - \sigma_{ij}^v n_i n_j$. For vertical cracks incised into a parallel-sided slab with $\sigma_{ij}^v$ given in dimensionless terms by (24),

$$\sigma_{ij}^v n_i n_j = \tau + z - s. \tag{31}$$

The dimensionless fluid pressure is

$$p_f = \max(r^{-1}(s - \eta - z), 0) \qquad\qquad \text{in a top crack,} \tag{32}$$
$$p_f = \max(-r^{-1}z, 0) \qquad\qquad \text{in a bottom crack,} \tag{33}$$

subject to

$$\beta = \int_{s-d_t}^{s-\eta} \max(w, 0)\mathrm{d}z \qquad \text{if satisfied for } \eta > 0, \qquad\qquad \eta = 0 \qquad \text{otherwise,} \tag{34}$$

while

$$\dot{d} = \max\left(K_{I,stat} - \kappa, 0\right). \tag{35}$$

## 3 Numerical Method

We use the displacement discontinuity boundary integral method as described in Crouch and Starfield (1983). To solve for the stress and displacement in $\Omega$, this method reduces the model to finding a vector-valued displacement discontinuity at the boundary $\partial\Omega$, from which stress, strain, and displacement fields can be computed through the use of a Green's function. Doing so requires us to introduce a fictitious elastic displacement field in the geometric complement of our domain, $\Omega' = \mathbb{R}^2 \setminus \Omega$, subject to the same stress boundary conditions as the original problem on the exterior part of the domain boundary $\partial\Omega$, and with stresses vanishing at infinity. Similarly, we use two copies of the boundary $\partial\Omega$ of our original bounded domain $\Omega \subset \mathbb{R}^2$, treating one copy as having an outward-pointing normal, the other as having an inward-pointing unit normal. On an exterior portion of $\partial\Omega$, the copy with an inward-pointing normal can be identified with the boundary of the fictitious exterior domain $\Omega' = \mathbb{R}^2 \setminus \Omega$. On any interior portion of $\partial\Omega$ (a crack), the copy with the inward-pointing normal is identified with the opposite side of the crack since displacement $u$ is naturally discontinuous at this region.

This procedure allows us to define the displacement discontinuity variable $\mathbf{D} = \mathbf{u}^+ - \mathbf{u}^-$, so $\mathbf{D} \cdot \mathbf{n}^+ = w$ in equation (9) for interior boundaries (cracks) as well external ones. This method proceeds by constructing the Green's function to relate normal and shear stress on the boundary to displacement discontinuity $D$ along the boundary (equation (A4)). Numerically, this is done by approximating $\partial\Omega$ as consisting of a finite set discrete straight line segments and computing the contribution to the Green's function for each, taking $D$ to be piecewise constant along each segment (or boundary element). We use a collocation method, forcing $\sigma_{nn}, \sigma_{nt}$ to take the imposed values at the center of these line segments; the contact conditions (10) imply that we may not have imposed values of normal stress everywhere, and we handle the resulting nonlinear complementarity problem





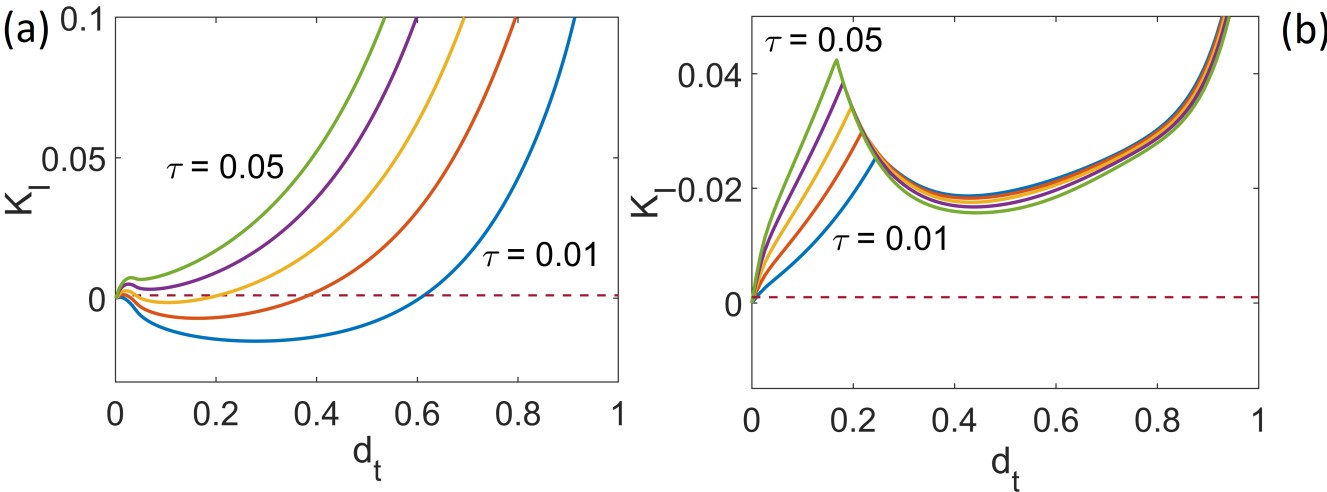

**Figure 2.** Scaled stress intensity factor versus the crack length for different values of scaled extensional stress $\tau = 0.01$ (blue), 0.02 (red) 0.03 (yellow), 0.04 (purple) and 0.05 (green), for (a) scaled water level depth $\eta = 0.04$ and (b) scaled water volume $\beta = 0.01$, without the contact condition. Fracture toughness $\kappa = 0.001$ is shown as maroon dashed line. Panel (a) is qualitatively equivalent to Figure 10 in van der Veen (1998a) and Extended Data Figure 5 in Lai et al. (2020).

by means of a semismooth Newton method (see appendix A). Once we have computed the displacement continuity solution, we calculate $K_I$ in terms of $D$ at the crack tip (equation (A7), see also Rice, 1968). We have conducted a number of tests on the boundary element code, computing known stress fields around simple crack configurations for which there are closed-form solutions, and the results for a single crack with simple prescriptions of water level and no constraint on crack width $w$ that 290 were previously reported in van der Veen (1998a,b) and Lai et al. (2020) using interpolated Green's functions (Tada et al., 2000).

## 4 Results

### 4.1 Dynamics of single crevasses: $K_I$ as a function of crack length and forcing parameters

In this paper, we consider a single crack of length $d_t$ incised vertically at the midpoint of the domain $x = W/2$, for a parallel-295 sided slab domain of unit thickness in dimensionless terms and a relatively wide domain width ($W = 10H$). The case of cracks simultaneously incised from the top and the bottom of the ice shelf will be dealt with in separate paper.

It is straightforward to see that the main forcing parameters are dimensionless extensional stress $\tau$, dimensionless fracture toughness $\kappa$, and either water table depth $\eta$ or water volume $\beta$, treating the density ratio $r$ as well as Poisson's ratio $\nu$ as constant. The dynamics of a single crack are simple: the crack will lengthen if the dimensionless stress intensity factor $K_I$ 300 corresponding to crack length $d_t$ and the given forcing parameters exceeds the dimensionless critical value $\kappa$.





Consider first a crack originating at the upper surface. Figure 2 shows the dimensionless stress intensity factor as a function of surface crack length, for different values of the tensile stress parameter $\tau$ that are plausible for floating ice shelves: (a) shows curves of $K_I(d_t)$ for different $\tau$ at a fixed water table depth $\eta = 0.04$ (for instance, a water level 20 m below the surface of a 500 m thick ice shelf) and (b) for a fixed scaled water volume $\beta = 0.01$ (this is equivalent to around 10 m$^2$ volume of water per

unit lateral width of a 500 m thick shelf with $E = 10^9$ Pa). In this Figure, for consistency with van der Veen's (1998a) approach, we suspend application of the contact constraint in equation (10) and instead impose the stress conditions $\sigma_{ij}n_j = -p_f n_i$ with an appropriate $p_f$ everywhere along the boundary. Note that we will see shortly that omitting contact constraint has significant dynamical consequences.

In each panel, the horizontal dashed line indicates an assumed scaled value of $\kappa = K_{Ic}/(\rho_i g H^{3/2}) = 0.001$ (which corre-

sponds to $K_{Ic} = 0.1$ MPa m$^{1/2}$ for a 500 m thick shelf). Points of intersection between the coloured curves of $K_I(d_t)$ and the dashed line are steady state crack lengths. More precisely, they are the end points of a finite region of steady states, since any crack length for which $K_I < \kappa$ is also a steady state. These end-point steady states are stable if $\partial K_I/\partial d_t < 0$ at the point of intersection (so a lengthening of the crack causes the stress intensity factor to decrease) and unstable if $\partial K_I/\partial d_t > 0$.

We see a fundamental difference in behaviour between the fixed water level and fixed water volume cases here. For fixed

water level (Figure 2a), we reproduce van der Veen's (1998a) results, at least qualitatively (there are minor differences between our models in terms of the assumed ice density): for a finite water level depth $\eta$, $K_I(d_t)$ at first increases with crack length $d_t$ at a rate proportional to $\tau d_t^{1/2}$ (see Weertman (1980) and appendix B) due to a dominant contribution from a constant extensional viscous pre-stress, and then decreases due to increasing cryostatic confining stress at depth. Once the crack tip is sufficiently below the prescribed water table depth $\eta$, water pressure in the crack increases at a larger rate with $d_t$ than overburden, and

therefore reduces the effect of cryostatic confining pressure. The result is that there are up to three points of intersection between the coloured curves and the dashed line, where $K_I(d_t) = \kappa$: two shallow cracks configuration with relatively small $d_t$, and one with large $d_t$. The shorter of the two shallow cracks is destabilized by the extensional stress and the second stabilized by increasing cryostatic pressure. The third, deep crack configuration is destabilized by water pressure increasing with depth, and the effect of the torque generated by viscous pre-stress and fluid pressure becoming more concentrated as the remaining

uncracked ice below the crack tip becomes thinner (see appendix B).

Since these points of intersection between the coloured curves and the dashed line define the end points of regions of steady states, we see that for intermediate $\tau$ there are two separate such regions: a region of very shallow cracks, for which the stress intensity factor, dominated by extensional stress $\tau$, is not yet large enough; and a larger one in which cryostatic pressure stabilizes the crack and water pressure is not yet large enough to destabilize it yet. The extent of these two regions depends

on the value of $\tau$: for small $\tau = 0.01$, the two regions merge, as the extensional stress is not large enough to cause $K_I$ to rise above $\kappa$, while for $\tau = 0.04$ and above, the larger region has become extinct as cryostatic stress no longer suffices to depress $K_I$. Importantly, Figure 2 shows these ranges for a single water table depth $\eta$, and we will explore the dependence of steady states on $\eta$ more systematically in subsection 4.4.

For the fixed water volume case in panel (Figure 2b), the stress intensity factor $K_I$ initially increases with crack length $d_t$,

as it does for the fixed water level case, before decreasing again. The initial increase in $K_I$ with $d_t$ is driven by the viscous


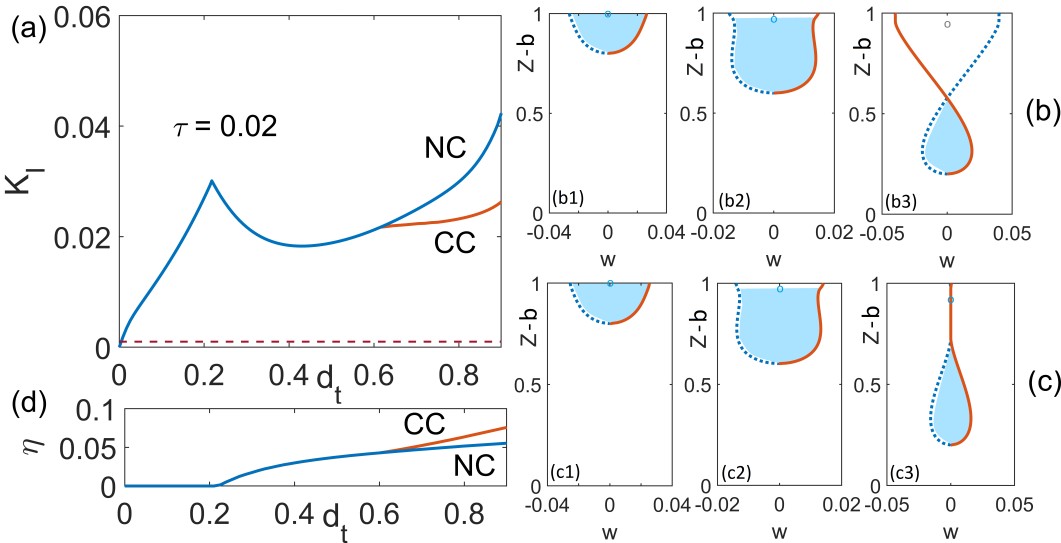

**Figure 3.** (a) Scaled stress intensity factor $K_I$ plotted against the crack length, $d_t$ for scaled water volume, $\beta = 0.01$, and scaled extensional stress, $\tau = 0.02$ with (blue) and without (orange) applying the contact condition, as indicated by CC and NC, respectively. (b) and (c): The corresponding crack width $w(z)$ plotted against $z$ for $d_t = 0.2$ (b1,c1), 0.4 (b2,c2) and 0.8 (b3,c3), without (row b) and with (row c) contact conditions. The left-hand crack face is shown in dotted blue, the right in solid orange, and water-filled parts of the crack in light blue. A circle indicates 'water level' (as defined by $\eta$) even where the crack is closed Note that the horizontal axis is scaled differently in each column for clarity. (d) Corresponding water level against crack length for the case shown in panel (a).

pre-stress $\tau$, and by increasing water depth in the crack driving up the fluid pressure $p_f$ near the crack tip: for small $d_t$, not all the prescribed water volume $\beta$ can be accommodated in the crack, and the water level remains at the ice surface with $\eta = 0$. Water depth and water pressure therefore increase initially with crack length $d_t$. This ceases to be the case once the crack is long enough to accommodate the prescribed volume, at which point water level then starts to decreases with further lengthening of

the crack, reducing the fluid pressure near the top of the crack and hence $K_I$; the point where all the required water volume $\beta$ is accommodated in the crack is easily identifiable by the discontinuity in $\partial K_I / \partial d_t$ in Figure 2b.

For larger $d_t$ beyond about 0.5, $K_I$ increases again. This can be attributed to the water level dropping more slowly (that is, $\eta$ increasing) with $d_t$ as the crack gets longer and the crack width is more limited due to increasing cryostatic pressure dominating the effect of the extensional stress $\tau$. The more limited width requires a greater water column height (Figure 3d),

which (perhaps counterintuitively) leads to an increasing stress intensity factor due to the more rapid increase of water pressure $p_f$ relative to crystatic pressure with depth.

For the chosen parameter settings in Figure 2b, we see that there is a single range of steady state crack lengths, at very shallow depths. As we will explore later, the dependence of $K_I$ on $d_t$ is in fact highly dependent on forcing parameters, and it is possible to generate steady states with much longer $d_t$. Before we do so, we address the role played by contact constraints

in determining $K_I$.





## 4.2 The effect of contact constraints

The results in Figure 2 were computed without imposing a contact constraint. As the crack gets longer, the crack walls in the upper portion of the crack bulge inwards towards each other. Without the constraint on crack width $w$ for the results shown in Figure 2, in van der Veen (1998a,b), and Lai et al. (2020), the crack walls not only touch but eventually overlap at larger

$d_t$ (Figures 4b) and 3b. That is, of course, aphysical. Figure 4a shows how re-introducing the constraint $w \geq 0$ affects the dependence of $K_I$ on crack length $d_t$. The most obvious feature of the orange curve (computed with the constraint in place) is that $K_I$ no longer becomes negative: a negative stress intensity factor invariably corresponds to negative crack width near the crack tip, and is therefore aphysical.

     The contact constraint however does not simply amount to setting $K_I$ to zero where the unconstrained solution predicts

negative $K_I$: the two solutions can differ from each other even where both predict $K_I > 0$ (for instance in Figure 4a for values of $d_t$ between 0.4 and 0.5). This difference occurs because contact between the crack faces can occur at higher elevations in the crack, as illustrated in Figure 4c2. This situation raises an interesting question about the water pocket that exists below the contact area: if water level is fixed at some depth $\eta$ below the surface and the contact area is at or below that elevation, how should water pressure in the deeper water pocket be prescribed? Our model *assumes* that fluid pressure continues to follow

a hydrostatic increase below the imposed water table depth even when the two are separated by a contact area, implying that asperities in the crack continue to provide a hydraulic connection across the contact area. That, however, is only one possible explanation, and isolated deeper water pockets (separated from the water table by a contact area) can conceivably behave as fixed water volumes instead.

     Generally, for fixed water table depth $\eta$, the effect of crack wall contact is to create a compressive normal stress greater than

the fluid pressure $p_f$ that would otherwise act in the crack. Overall force and torque balance dictate that tensile stresses and torques in the unbroken portion of ice below the crack tip (for $b < z < s - d_t$) increase to compensate. This explains why the stress intensity factor increases in Figure 4 when the contact constraint is imposed.

     The effect of the contact constraint on the stress intensity factor differs for the constant volume case. With the parameter values used in Figure 3a, $K_I$ is positive for all $d_t > 0$, no matter whether the contact constraint is applied or not. However,

the solution with the contact constraint applied has a lower $K_I$ than the solution without once a contact area forms higher up in the ice (above $d_t > 0.6$). The reason for this is that, with a contact condition, a longer water pocket (with $w > 0$) naturally forms (Figures 3c3 versus 3b3) and a lower water pressure (or equally, a larger $\eta$) suffices to accommodate the prescribed water volume (Figure 3d).

## 4.3 Larger extensional stresses and lower water levels

Figures 2, 3 and 4 were computed for relatively small values of $\tau$, and equally for a small water depth $\eta$ and water volume $\beta$, and the qualitative behaviour of $K_I(d_t)$ is much the same as in van der Veen (1998a). Note that the behaviour of $K_I(d_t)$ changes substantially for larger $\tau$ as shown in Figure 5, which is computed with the contact constraint in place. In Figure 2, there is an increase in $K_I$ with $d_t$ at small crack lengths due to the dominant action of the extensional stress $\tau$. $K_I$ then reaches

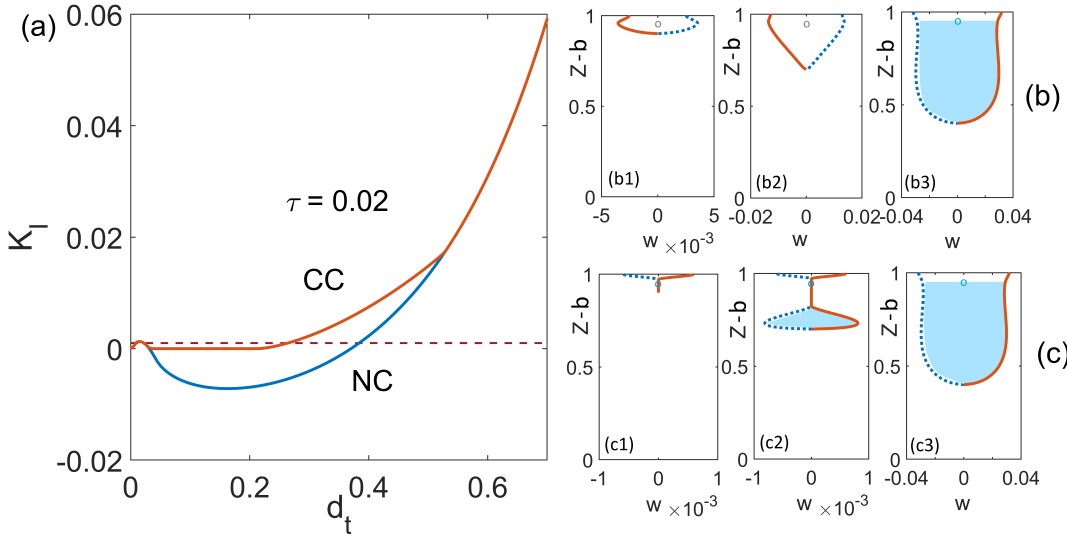

**Figure 4.** (a) The equivalent of Figure 3 for the case of fixed water level $\eta$, with parameter values $\eta = 0.04$ and $\tau = 0.02$, using the same plotting scheme as Figure 2. Rows (b) and (c) show crack opening and water filled parts of the cracks for $d_t = 0.1$ (b1,c1), 0.3 (b2,c2) and 0.6 (b3,c3).

a maximum at small to moderate $d_t$, at a point where cryostatic pressure starts to dominate normal stresses on the crack. In both
panels of Figure 2, there is a local minimum in $K_I$, and the stress intensity factor increases again due to rising fluid pressure for cracks that span most of the ice thickness. By contrast, for larger $\tau$, the maximum in $K_I$ is reached at significantly greater depths, and $K_I$ may or may not increase as $d_t \rightarrow 1$.

     In fact, Figure 5 indicates that the generic behaviour for a crack that spans nearly the full ice thickness is that either $K_I = 0$ or $K_I \rightarrow +\infty$ as $d_t \rightarrow 1$. This can be attributed to the torques generated by extensional stress $\tau$, cryostatic pressure $s - z$ and
fluid pressure $p_f$ (or contact stress in the contact areas) on the remnant "neck" of ice that still connects the two sides of the domain, with a singular $K_I$ favoured by smaller water level depth $\eta$. We explain this in greater detail in appendix B, where we show how it is often possible to determine the critical combination of parameter values at which the change in $K_I$ from zero to infinite occurs.

     As in section 4.1, we can read steady state ranges off these plots by identifying where $K_I < \kappa$. For the fixed water level
case, we see that there is still a short region of shallow steady state cracks near $d_t = 0$, and potentially a region of much larger steady states. Whether that latter region exists, and whether it extends all the way to $d_t = 1$ or terminates at an unstable steady state, depends on parameters: For smaller $\eta$ and $\tau$, we still obtain a region of steady states that terminates short of the full ice thickness $d_t = 1$, while for larger $\tau$, we either obtain a region that extends to $d_t$ if $\eta$ is sufficiently large (that is, water pressure is lower), or none at all at smaller $\eta$ (higher water pressure). Note that an increase in either $\tau$ or reduction in $\eta$ always shrinks
the region of steady states, as might be expected on physical grounds.

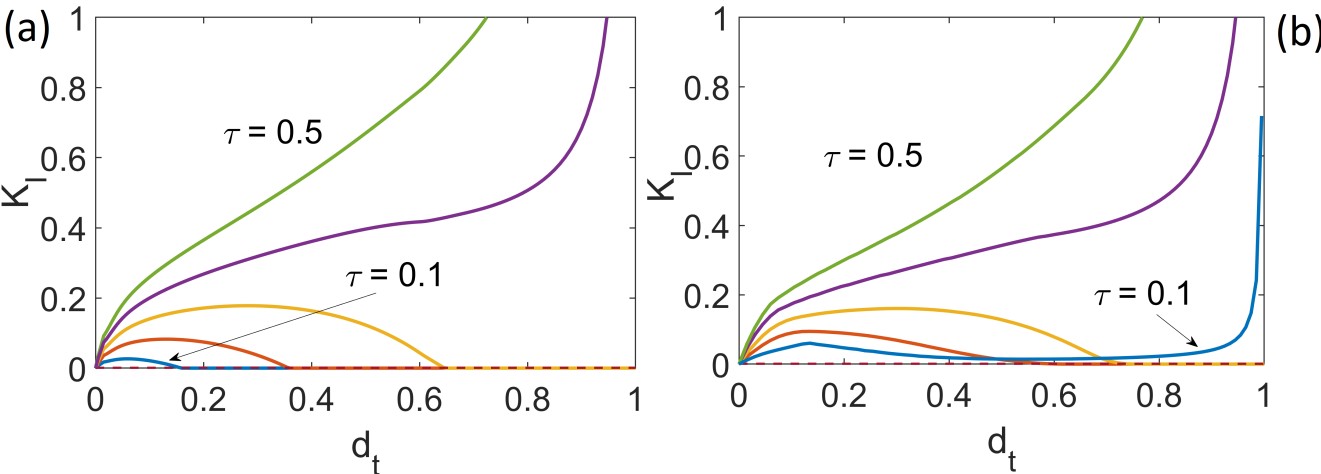

**Figure 5.** (a) Stress intensity factor versus the crack length for large scaled extensional stress, $\tau = 0.1$ (blue), 0.2 (orange) 0.3 (yellow) 0.4 (purple), 0.5 (green) for (a) $\eta = 0.6$ and (b) $\beta = 0.01$, both computed with the contact constraints being satisfied.

For fixed water volume, we see a region of larger steady states appear at intermediate values of $\tau$ where there was none for small $\tau$ (Figure 5b). The appearance of these steady states corresponds to the plots $K_I(d_t)$ for $\tau = 0.2$ and 0.3 having $K_I = 0$ and lying below the curve for $\tau = 0.1$ at larger $d_t$ in Figure 5b. This is a major difference between the two hydrology models, since increased $\tau$ invariably reduces the range of steady states for the fixed water level case. Physically, the behaviour for fixed water volume $\beta$ can be explained by larger extensional stress opening the crack further, so the prescribed water volume can be accommodated lower down in the ice column, leading to reduced fluid pressure and therefore to lower $K_I$.

### 4.4 Dependence of steady states on parameters

We have so far focused on identifying steady states for discrete parameter values of $\tau$ and $\eta$ as shown in Figures 4 (or 2, if we ignore the contact constraint) and 5. Instead, we can also plot the boundaries of the region of steady states explicitly as functions of the model parameters: such a plot is effectively a bifurcation diagram for the semi-smooth dynamical system defined by equation (20).

Figure 6a,b shows an example for a number of fixed water level values, using the dimensionless extensional stress $\tau$ as the parameter being varied. The area to the left of each curve (shaded in light grey for $\eta = 0.06$ in panel a, and for $\eta = 0.2$ in panel b) represents steady state values of $d_t$ for the corresponding $\tau$. The boundary curves of these areas are computed by solving for $K_I = \kappa$ using a continuation method, with $K_I$ computed from the model with contact constraint enabled. A dashed portion of the boundary curve corresponds to an unstable boundary as defined above, a solid curve to a stable boundary. Outside of the steady state regions, $d_t$ invariably increases with time (since, by construction, crack length cannot shrink).

The results in Figure 6a,b correspond to the observations we have made previously based on Figures 4 and 5a: In panel a, we see the split of the steady state area into two separate parts at relatively small $\tau$ and $\eta$. The lower region (near $d_t = 0$) once more



represents very shallow steady state cracks. This regions thins progressively with increasing $\tau$ as $\tau^{-2}$ (since $\kappa = K_I \sim \tau d_t^{1/2}$ on the dashed upper boundary of this lower region, see appendix B) but never disappears entirely: provided an initial crack is shallow enough, it may never grow at all.

Note that the upper boundary of this lower region is insensitive to water level $\eta$. The lower dashed boundary curve may appear to exist only for $\eta = 1$ as indicated by the maroon curve. That is however simply the result of the boundary curves for

different water level values being indistinguishable. In most cases, they coincide exactly because they correspond to dry cracks with $d_t < \eta$. The case $\eta = 0$ (a completely full surface crevasse) is something of an exception: here $d_t > \eta$ everywhere and the boundary curve is not identical to the others, but close enough almost everywhere to be indistinguishable as $K_I$ is dominated by the effects of the pre-stress $\tau$. By contrast with the case of non-zero water level depth, there is also no split into a lower and an upper steady state region for $\eta = 0$; only the narrow band of shallow steady states exists. Note that this reproduces the results

in Lai et al. (2020), and in particular the corresponding curves for water-filled crevasses in Figures 5c,d of the supplementary material to Lai et al. (2020)

For non-zero water level depth $\eta$, there is often an upper region consisting of steady cracks of more substantial size. This region is enlarged for a fixed extensional stress $\tau$ if water level drops ($\eta$ increases), and shrinks if $\tau$ is increased for fixed $\eta$. In fact, the upper region only exists conditionally, for sufficiently small $\tau$ (given $\eta$) or sufficiently large $\eta$ (given $\tau$). This is to be

expected: high water levels (small $\eta$) or large extensional stresses $\tau$ will destabilize large cracks and cause them to propagate all the way through the ice.

In greater detail, depending on $\eta$, there are two possible configurations for the upper region of steady states, and two possible ways in which it shrinks and eventually disappears as $\tau$ is increased at fixed $\eta$. At small to moderate $\eta$, the larger range of steady states lies between a lower, stable boundary (marked by a solid line) and an upper, unstable boundary (dashed line), with the

two meeting at a turning point for some critical value of $\tau$; this turning point corresponds to a so-called saddle-node bifurcation of the dynamical system (20) (albeit a somewhat unusual one because the system is semi-smooth, and the entire region between upper, unstable and lower, stable curve consists of steady states).

By contrast, for sufficiently low water levels (large $\eta$), the upper region of steady states lies between a lower, stable boundary (solid line) and the maximum possible crack length of $d_t = 1$ as the upper boundary: arbitrarily thin necks of ice (with $d_t$

arbitrarily close to the full ice thickness value of 1) can remain in steady state. As $\tau$ increases, the lower boundary simply approaches $d_t = 1$ smoothly, and the upper region of steady states disappears when the two meet at finite $\tau$ (see appendix B).

The constant water volume case differs considerably from the constant water level case. In Figure 6c, we show an analogous bifurcation diagram for the system at different fixed values of $\beta$, with $\tau$ again being the parameter allowed to vary continuously. As already deduced from Figures 3 and 5b, we see again a narrow region of steady states close to $d_t = 0$ (shaded in grey, with

a dashed, unstable upper boundary). For non-zero water volume $\beta$, the boundary curve of that region is in fact identical to that for $\eta = 0$, since these solutions correspond to full surface cracks that are unable to accommodate all of the prescribed water volume.

For a non-zero water volume, an upper region of steady states does appear (shaded in grey for $\beta = 0.005$), but only *above* some critical $\tau$: as previously discussed, the opening of the crevasse at larger extensional pre-stress allows the prescribed water


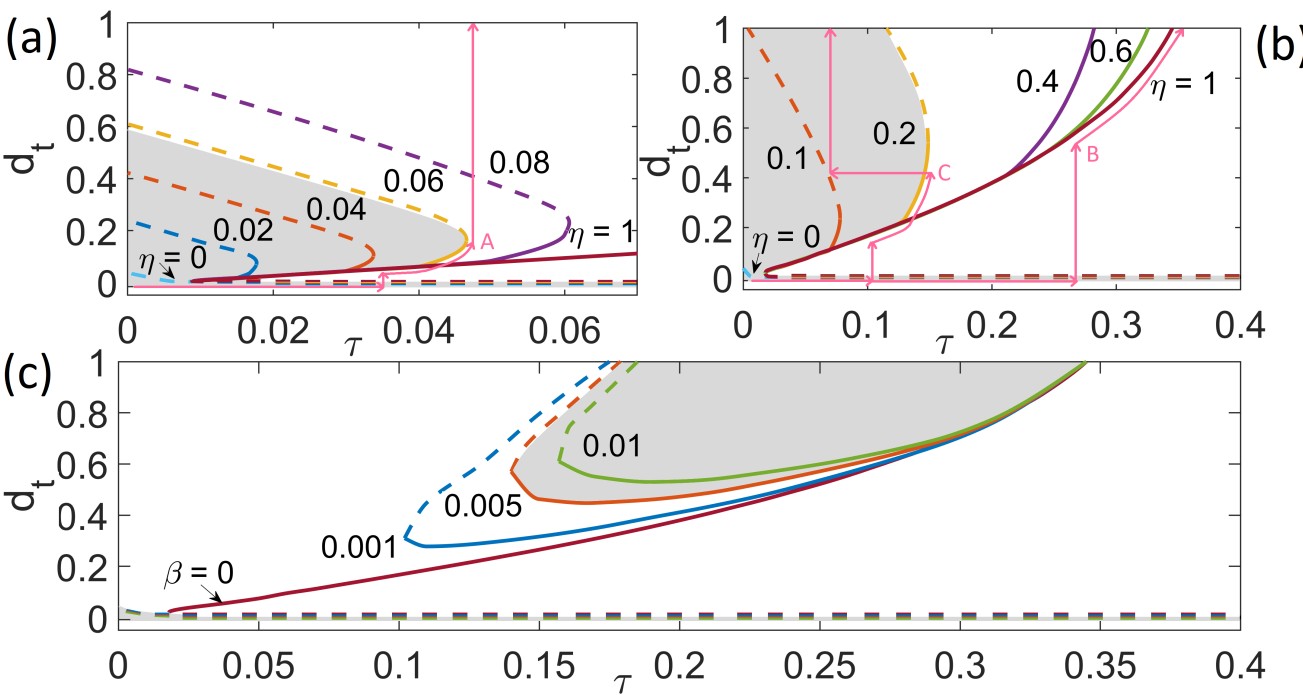

**Figure 6.** Bifurcation diagram with contact condition, for an ice shelf with a single crack at the top. Each heavy solid or dashed curve is the locus of points satisfying $K_1(d_t; \tau, \eta) = \kappa$ or $K_1(d; \tau, \beta) = \kappa$) in the $(\tau, d_t)$-plane at a fixed value of $\eta$ or $\beta$ as indicated, representing the boundary of a region of steady states in which $K_1 \leq \kappa$. A solid portion of the boundary curve indicates a stable boundary, unstable if dashed. (a) and (b): the case of constant $\eta$, the value of $\eta$ being indicated for each curve. The region of steady state generally lies to the left of each curve, as indicated by gey shading for $\eta = 0.04$ (panel a) and $\eta = 0.2$ (panel b). The two panels differ in the range of $\tau$ and $\eta$. Note that there is a narrow region of steady states near $d_t = 0$ in each panel (elevated slightly above the bottom of the plot for visibility). Also shown as narrow pink curves are three phase paths that the point $(\tau, d_t)$ can follow under changes in $\tau$ and $\eta$. All paths begin at a small $d_t$ and $\tau = 0$, paths A and B correspond to monotonic increases in $\tau$ at different fixed values of $\eta = 0.06$ (A) and 1 (B). Path C corresponds to a monotonic increase and subsequent decrease in $\tau$ at $\eta = 0.2$ followed $\tau$ being held fixed while $\eta$ is lowered to 0.1, (C) The case of constant $\beta$, values as indicated, steady state region indicated for $\beta = 0.005$ using the same plotting scheme as panels a and b. Note that there is again a narrow region of steady states near $d_t = 0$.





volume to be accommodated at greater depths, leading to perhaps counterintuitively reduced stress intensity factor values $K_I$. The larger the prescribed water value, the smaller the upper region of steady states becomes, and the values of $\tau$ at which we find the upper region generally exceed the range $\tau \leq (1-r)/2 = 0.05$ that is typically expected for ice shelves. For completeness, note that the upper region of steady states becomes much more extensive as water volume $\beta$ shrinks: the case $\beta = 0$ (a dry crack) becomes identical to the dry crack case $\eta = 1$ in Figure 6b.

The work above has focused on surface cracks. The case of a single basal crack (van der Veen, 1998b) is in fact simpler than the surface crack, since there is no hydrological parameter $\beta$ or $\eta$ to take care of. The dependence of $K_I$ on crack length $d_b$ is again shown in Figure 7. Note that contact areas never form lower in the crack, but are always adjacent to the crack tip (Figure 8a), and the effect of introducing the contact constraint is simply to truncate $K_I$ computed from the model without a contact constraint at zero (compare Figure 7a and b). As in Figure 5, we see a pattern of $K_I$ increasing from $K_I = 0$ at $d_b = 0$.
For $\tau$ less than a numerically determined critical value of $\tau_{crit} = 0.039$, $K_I$ then decreases again and vanishes for $d_b$ close to unity, while for $\tau$ above this value, $K_I$ diverges to $+\infty$ as $d_b \to 1$ (see Figure 8c); this can again be attributed to the torque exerted on the narrowing neck of ice connecting the two halves of the domain as described in appendix B: there we show that the critical value for $\tau$ at which this occurs can in fact be explained by the torque changing signs, with a theoretical calculation giving a value of $\tau_{crit} = \{r^{-1}[1 - (1-r)^3] - 1\}/3 = 0.0367$. We attribute the difference between the value to the difficulty in
computing stresses accurately at finite element sizes when the remnant neck of ice is shrunk towards zero size.

The resulting dependence of the region of steady states on the parameter $\tau$ is shown in Figure 8b: as for the surface crack, there is always a narrow range of shallow steady states near $d_b = 0$, and generally a larger range of steady states extending all the way to $d_b = 1$ (corresponding to full crack penetration). That latter range becomes narrower and vanishes at the critical $\tau = 0.039$. Because that vanishing corresponds to the behaviour of $K_I$ near $d_t$ flipping from $K_I = 0$ ($K_I \to -\infty$ when there is no
contact condition) to $K_I \to +\infty$, the critical value is independent of $\kappa$. Note that the same result is implicit in Figure 4c of the extended data in Lai et al. (2020), where complete penetration of a basal crack is also displaced as occurring at a fixed value of $\tau$ close to 0.039.

## 4.5 Calving

We can understand calving as the formation of a crack that spans the entire thickness of the ice as the result of a change in the
forcing applied to the ice shelf. In order to make sense of that using only the model in the present paper, we have to assume not only that we can use the parameters $\tau$, $\eta$ and $\kappa$ to represent changes in forcing, but also that we can ignore changes in the specific form of pre-stress $\sigma_{ij}^v$ given in (24) (through which $\tau$ is ultimately defined) as well as changes in the local ice shelf geometry into which a new crack is incised, or in which an existing crack is lengthened. These assumptions of course do not hold in practice: viscous pre-stress evolves over a single Maxwell time (given by the ratio ice viscosity to Young's modulus,
typically hours to a day for a polar ice shelf) once an initial crack has formed , and will not remain of the form (24). Secondly, the local geometry of ice shelf will also evolve, although more slowly, once a partial crack (which does not span the entire ice thickness) has been incised (Yu et al., 2017).



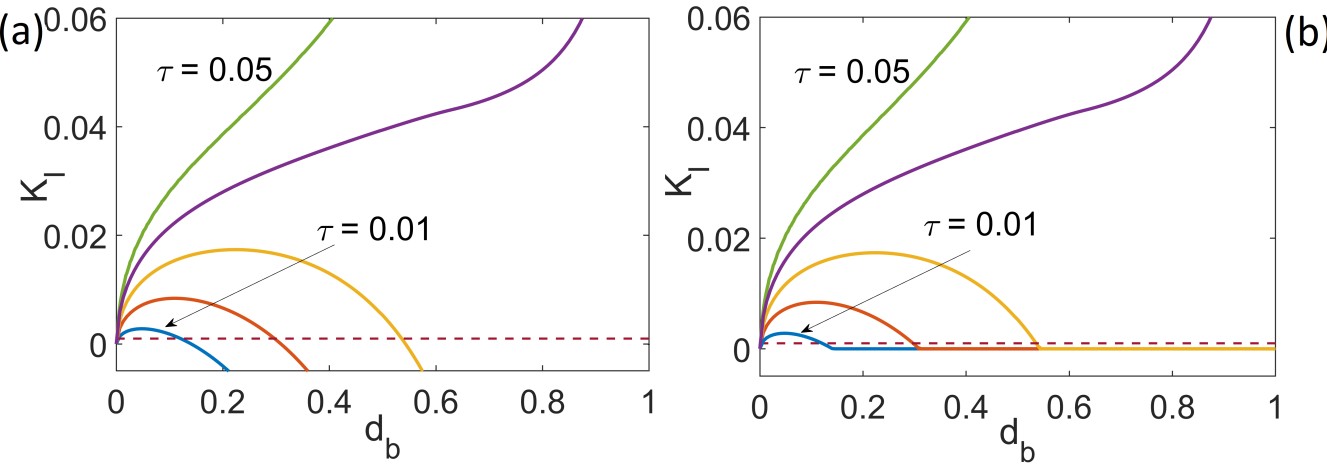

**Figure 7.** (a) Stress intensity factor against crack length for a basal crack for different values of $\tau$ (same plotting scheme as Figure 2, without (a) and with (b) contact conditions. Panel (a) is qualitatively equivalent to Figure 2 in van der Veen (1998b).

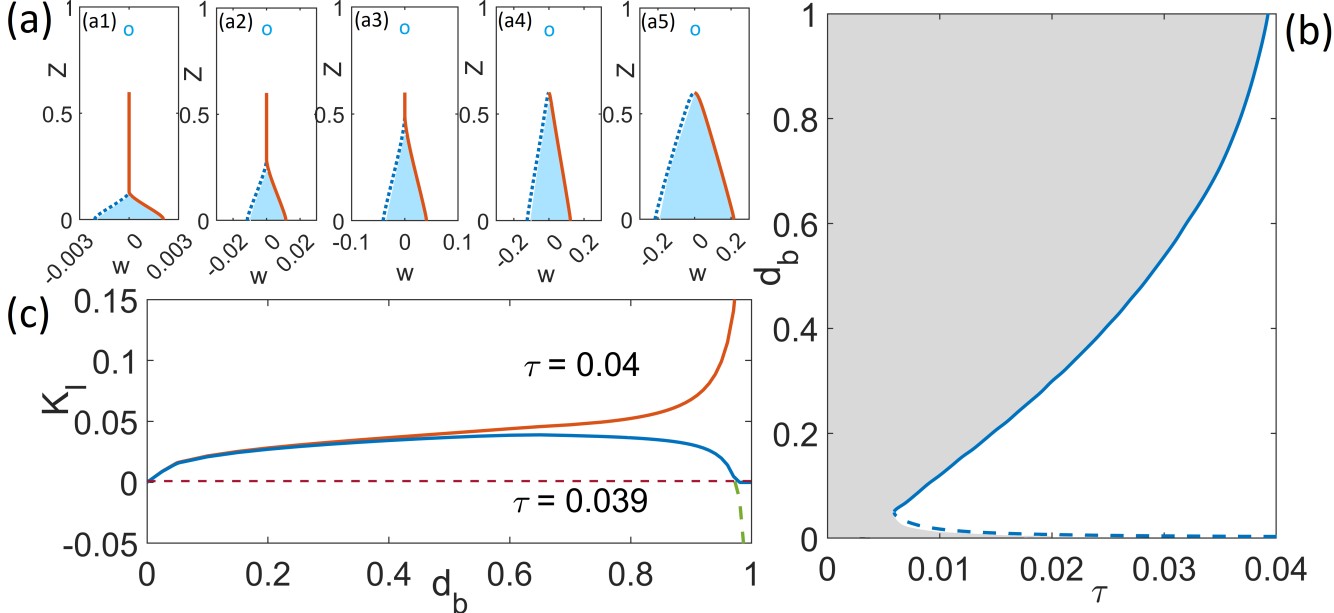

**Figure 8.** (a) Crack opening $w(z)$ computed with contact condition, at $d_b = 0.6$ for $\tau = 0.01$ (a1), 0.02 (a2), 0.03 (a3), 0.04 (a4) 0.05 (a5), same plotting scheme as Figure 3c; water level as indicated by the circle is sea level. (b) Bifurcation diagram for the basal crack for $\kappa = 0.001$, same plotting scheme as Figure 6 except that there is now a single boundary curve since there is no hydrological parameter. (c) Stress intensity factor versus basal crack length for $\tau = 0.039$ and 0.04. The green dotted line shows the result for $\tau = 0.039$ without the contact condition. Calving occurs when $K_I$ changes (under changes in $\tau$ from being zero to diverging to $+\infty$ near $d_b = 1$, and is independent the stress intensity factor as a result.



It is however instructive to pursue the evolution of cracks as described by our model alone under changes in parameters, leaving the evolution of viscous pre-stress and geometry to future work. Doing so allows us to develop a framework for
understanding calving. Under the assumptions we are making, the parameters $\tau$, $\eta$ and $\kappa$ can be expected to evolve slowly compared with the dynamic crack propagation time scale. That time scale is inertial in our model, but the same would be true if we were to use a hydrofracture model with a dynamically evolving crack fluid pressure field $p_f$ (Spence and Sharpe, 1985)). The stress parameter $\tau$ represents deviatoric stresses in the ice, and evolves as the large scale ice geometry does (over many years due to ice flow, unless there is another calving event happening elsewhere in the shelf), while water level $\eta$ is likely to
evolve seasonally. The scaled fracture toughness $\kappa$ invariably changes slowly: with fixed $K_{Ic}$, $\kappa$ changes purely because ice thickness evolves.

Following the discussion above, consider an example of how slow changes in the parameter $\tau$ can lead to calving. In Figure 6a, the point $(\tau, d_t)$ is a possible steady state crack configuration for given $(\eta, \kappa)$ if it is inside the region of steady states or if it is on a solid (stable) part of the boundary. If the point lies initially inside the steady state region rather than on its boundary, then
the value of $d_t$ will not change as a result of a parameter change: under a change in $\tau$, the point in question moves horizontally (parallel to the $\tau$-axis). Suppose the point reaches a stable part of the boundary under such a change in $\tau$ (that change being, invariably, an increase). Then any further increase in $\tau$ would take $(\tau, d_t)$ outside of the steady state region if $d_t$ were to remain unchanged. Of course, instead of exiting the steady state region, $(\tau, d_t)$ will simply follow the stable boundary while this is possible.

By contrast, if the point reaches the end of the stable part of the boundary (as it does at the saddle-node bifurcation in Figure 6), or if the point otherwise reaches an unstable part of the boundary of the shaded region (as is possible under changes in $\tau$ with the right initial crack length $d_t$), or if it starts outside the region altogether, then $d_t$ will increase rapidly while the parameter value $\tau$ remains essentially constant (since we assume that forcing changes on much longer time scales than that associated with crack propagation). In other words, the point $(\tau, d_t)$ moves vertically in the bifurcation diagram, parallel to the $d_t$-axis. It
will continue to move in that direction until it hits a stable boundary of a region of steady states (which is possible if the crack starts in the narrow region of shallow steady states and transitions to a separate region of larger steady states above) or until it hits the line $d_t = 1$ and the domain is severed. The latter represents a calving event (as illustrated by path (A) in Figure 6a). Here, crack length does not continuously grow to span the entire ice thickness, but becomes unstable and rapidly propagates the remaining ice thickness.

An alternative calving mechanism for lower water levels (larger $\eta$) is that there is no saddle-node bifurcation, and the stable boundary of the region of steady states smoothly approaches $d_t = 1$ under increases in $\tau$ (see Figure 6b for $\eta = 0.4$, 0.6 1). Calving in this fashion simply involves crack length growing continuously until the last remaining neck of ice is severed (see path (B) in Figure 6b), although this requires not only low water levels but also unrealistically larger extensional stresses for the surface cracks in Figure 6b (recall once more that in an unconfined ice shelf, $\tau \approx 0.05$, and we expect lower extensional
stresses in typical buttressed ice shelves).

Calving can also occur through changes in $\eta$, $\kappa$, or multiple parameters at once. At fixed $\tau$, calving due to decreases in $\eta$ can occur in a similar fashion to calving due to increases $\tau$: either through reaching a saddle-node bifurcation or the unstable





boundary of the upper region of steady states for small to moderate $\tau$, or through the lower stable boundary of that upper region smoothly reaching $d_t = 1$. Combined changes in $\tau$ and $\eta$ can further complicate the style of calving, and make it more likely

that calving occurs by reaching the upper, unstable boundary of the upper region of steady states as illustrated in Figure 6a: in particular a temporary increase in $\tau$ that does not in itself induce calving may still lengthen the crack appreciably, which a subsequent reduction in $\tau$ does then not reverse. If $\eta$ is reduced later (that is, the water level rises), the previously lengthened crack may become unstable and cause calving even if the initially much shorter crack would not be susceptible to calving at the same combination of $(\tau, \eta)$ (see path (C) in Figure 6b).

As discussed at the end of the previous section, the calving mechanism for bottom cracks is analogous to the second mechanism for surface cracks described above: calving occurs when the upper, stable branch of the boundary curve of the stable region in Figure 8b, reaches $d_t = 1$.

For the fixed water volume case, the situation appears quite different: for most of the water volume values in Figure 6c, except for exotic initial conditions, calving at realistic values of extensional stress $\tau$ appears to occur whenever $(\tau, d_t)$ hits the

unstable boundary of the lower region of steady states, consisting of very shallow cracks. It seems implausible that limited amounts of water should cause calving at low stress but not at high stress. We discuss this further in section 6, where we argue that Figure 6c may in fact be a red herring as a description of calving if viewed in a three-dimensional geometry.

## 5 Calving Law

A calving law is a parameterization of calving that can be used in a large scale ice sheet model. Ideally, we would like something

like a relationship between the different model parameters. For the case of a bottom crack propagating upwards, the results at the end of section 4.4 (see Figure 8) in fact furnish a calving law, of the very simple form

$$f_b(\tau) = \tau - \tau_{crit} = \frac{R_{xx}}{\rho_i g H} - \tau_{crit} = 0, \tag{36}$$

since the critical value $\tau_{crit}$ does not depend on scaled fracture toughness and there is no hydrological parameter to take care of. Numerically, we have found $\tau_{crit} = 0.039$, while theory (appendix B) predicts a slightly lower value of $\tau_{crit} = 0.039$. An

analogous result is shown in Figure 4c of the extended data in Lai et al. (2020).

For a prescribed surface water level, an equivalent relationship would take the form the form $f_c(\tau, \eta, \kappa) = 0$ *at which* calving happens in the sense of $d_t$ rapidly transitioning to a value of one when that relationship is satisfied, or reaching unity continuously. If there were such a relationship, then using the definition of $\tau$, $\eta$ and $\kappa$, calving for a surface crack with fixed water level would occur when

$$f_t\left(\frac{R_{xx}}{\rho_i g H}, \frac{h_w}{H}, \frac{K_{Ic}}{\rho_i g H^{1/2}}\right) = 0. \tag{37}$$

These could be implemented in a large-scale ice sheet model, where thickness and stress are dynamical variables, and a surface hydrology model could conceivably be developed to predict water level $h_w$. In fact, structurally, these calving laws are analogous to others such as that in Nick et al. (2010) and Schoof et al. (2017), in which calving happens at a critical thickness $H$





that depends on extensional stress and a hydrological parameter analogous to $h_w$: equation (37) defines an implicit relationship between $H$ and the remaining model parameters.


The problem is however that there is no unique function $f_t$, in general. To understand why this is so, recall our discussion in section 4.5 of the sample phase paths 6: Take for instance path (C), which leads to calving through reaching the upper, unstable boundary of the upper region of steady states at a combination of $(\tau, \eta)$ for a crack that has not been lengthened and previously would remain steady. There is no unique calving behaviour even in the simple set-up considered here, and calving is strongly

history-dependent: cracks that were lengthened previously (by parameter changes that were subsequently reversed) favouring earlier calving.

In principle, we have a recipe for computing the trajectory of $d_t$ in phase space in response to slow changes in parameters $(\tau, \eta, \kappa)$: if $T$ is a slow time variable associated with large-scale evolution of the ice shelf, then the current crack length $d_t(T)$ can be computed in terms of the 'most recent' crack length $d_t^-(T) = \sup_{T' < T} d_t(T')$ and the set of stable steady states $S$,

defined through $S(\tau, \eta, \kappa) = \{d_t : K_I(d_t, \tau, \eta) < \kappa\} \cup \{d_t : K_I(d_t, \tau, \eta) = \kappa \text{ and } \partial K_I/\partial d_t < 0\} \cup \{1\}$, where sup and inf are the usual lowest upper bound and greatest lower bound, and we include the 'calved' solution $d_t = 1$ in the set of stable steady state. In this notation, we have

$$d_t(T) = \inf\{d_t' : d_t' \in S \text{ and } d_t' > d_t^-(T)\}, \tag{38}$$

and calving occurs when $d_t$ first reaches unity.

This is clumsy-looking but possible to implement numerically in a dynamical ice sheet model if the set $S$ is known a priori from offline computations such as those shown in Figure 6. We can construct a simpler description of calving, of the form (37), if we assume that crack length will always track the lower, stable boundary of the upper region of steady states in Figure 6, whenever that boundary exists. Within our model, this occurs principally if $\tau$ never decreases in time and $\eta$ never increases in time; alternatively, we can assume that the crevasse will heal rapidly (relative to the time scale over which the forcing

parameters change), resetting $d_t$ to the nearest shorter length at which $K_I$ is equal to the fracture toughness $\kappa$.

If crack length is forced in this way to follow that lower boundary whenever possible, then calving occurs either because the lower boundary terminates at a saddle-node bifurcation, or because it reaches the maximum possible crack length $d_t = 1$, The location of the bifurcation, or of the point at which the lower stable boundary reaches $d_t = 1$, defines $\tau$ as a function of $\kappa$ and $\eta$. More generally, it defines a surface in $(\tau, \eta, \kappa)$ space as in equation (37). As an example, Figure 9 for instance shows

$\tau$ at calving as a function of $\eta$ for the fixed values of $\kappa = 0$ to 0.01. Note that the curves corresponding to different $\kappa$ differ primarily through their starting points as shown in the inset Figure: the saddle-node bifurcation first appears at some finite $\tau$, which corresponds to the split into the lower and upper regions of steady states in Figure 6a, and that starting point depends on $\kappa$. For larger water depths $\eta$, calving becomes insensitive to the scaled fracture toughness.

In fact, for combinations of sufficiently large $\tau$ and $\eta$, calving occurs through the continuous shrinking of the 'neck' of ice

of thickness $1 - d_t$ as stress increases or water depth $\eta$ decreases, and calving is controlled by torques on that neck of ice. Appendix B furnishes an asymptotic form for $f_t$ in that case, of the form

$$f_t(\tau, \eta, \kappa) \sim \eta - 1 - [r(1 - 3\tau)]^{1/3}, \tag{39}$$

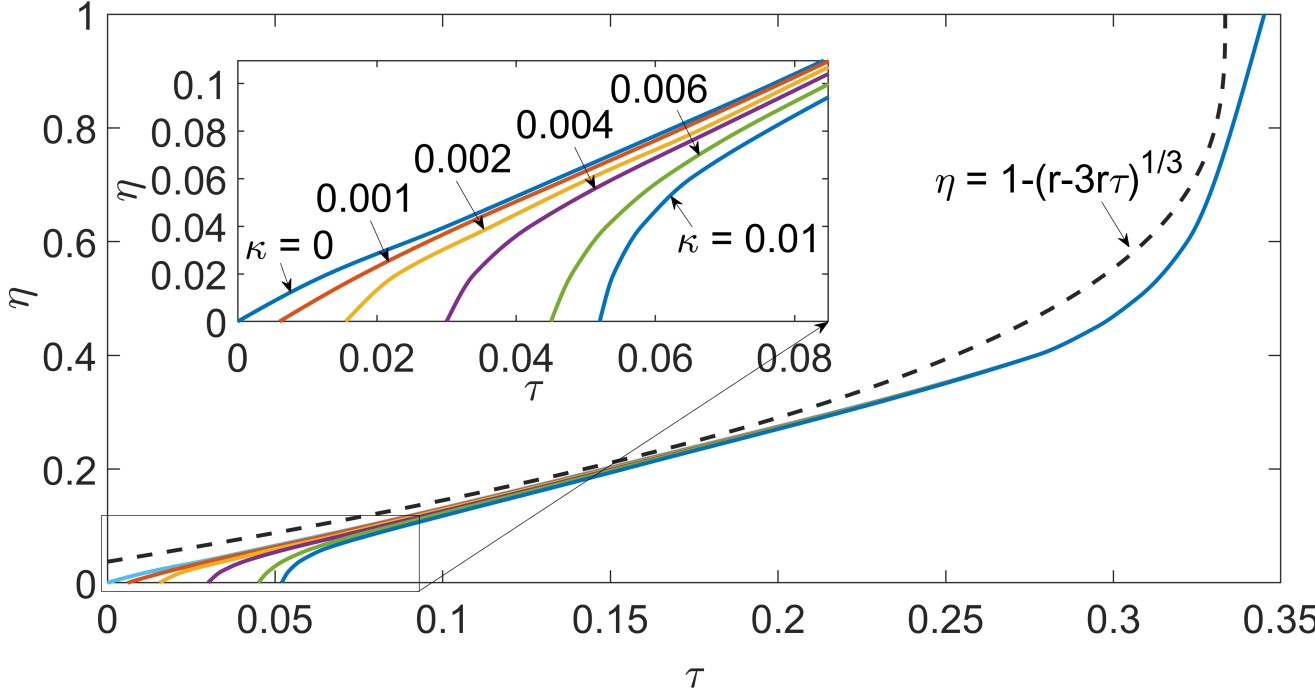

**Figure 9.** The location of the saddle-node bifurcation point in Figure 6 (if it exists), or the location where the boundary reaches $d_t = 1$ when there is no saddle-node bifurcation, plotted as $\tau$ at the bifurcation point against the corresponding $\eta$, for different values of scaled fracture toughness, $\kappa$. An enlargement of the bottom left-hand corner identifying the colour scheme is shown at top left. The dashed curve represents the analytical calving law in equation (39) of appendix B.

valid for $\tau$ close to a critical value of 1/3. This form of the calving law is plotted as a dashed line in Figure 9: it turns out to capture the calving law quite well for a relatively large range of $\tau$. As in the case of bottom cracks (see the end of section 4.5),

we find that theory and numerical results do not agree perfectly even where the theoretical result is expected to be accurate (near $\tau = 1/3$, $\eta = 1$). We attribute that once more to the difficulty in computing $K_I$ accurately using finite-sized boundary elements when the ice neck thickness $1 - d_t$ shrinks to zero.

As discussed at the end of section 4.5, the finite water volume case does not offer any obvious path to a calving law: fracture propagation can be expected to occur across the full ice thickness for realistic values of $\tau$ once $(\tau, d_t)$ reaches the unstable

upper boundary of the lower region of steady states in Figure 6c, and is as such heavily dependent on initial conditions. We expect no equivalent to (37) in that case. What is more, however, is that it also remains unclear whether that complete fracture propagation would necessarily correspond to calving, or perhaps instead simply to the drainage of the prescribed water volume through a localized slot that reaches the bottom of the ice shelf. We return to this shortly below.





# 6  Conclusions

In this paper, we have extended the two-dimensional theory for penetration of partially water-filled crevasses in an ice shelf by
linear elastic fracture mechanics as developed previously by van der Veen (1998a,b) and Lai et al. (2020); as in these papers, the
situation we have in mind is a crevasse that is distant from either grounding line or calving front (see also Hooke and Hanson,
2017; Wagner et al., 2016). We have explicitly formulated the crevasse penetration problem in terms of a viscous pre-stress
and an elastic stress induced by the sudden introduction of a crack (see also Yu et al., 2017), though the mathematical form

for a domain that takes the form of a parallel-sided slab prior to crack penetration and is identical to that in van der Veen
(1998a,b) and Lai et al. (2020) except for the addition of contact constraints: owing to a solution method based on interpolated
Green's functions being used in van der Veen (1998a,b) and Lai et al. (2020). These authors are not able to prevent crack faces
from intersecting each other aphysically when crack opening becomes negative. We are able to rectify that by modifying the
standard displacement discontinuity boundary element method (Crouch and Starfield, 1983), applicable to arbitrary domain

shapes, to account for contact constraints and solve for the crack tip stress intensity factor and crack propagation rate (section
3 and appendix A), based on a quasi-static crack propagation description due to Freund (1990).

The focus of our paper is to determine conditions under which a crack is able to propagate across the entire thickness of
the ice, which we interpret as a calving event. While van der Veen (1998a,b), using a nearly identical basic fracture mechanics
model (omitting only the contact constraints) largely stopped short of identifying conditions for calving this way, Lai et al.

(2020) do so for a single surface crack that is either completely dry or completely filled with water. Here, we extend their
work to allow for a more general prescription of hydrology: we consider either a water level at a prescribed depth below the
ice surface as in van der Veen (1998a) (mimicking a water table in an englacial drainage system) or a prescribed, fixed water
volume in the crack as a more physics-based generalization of the fixed water column height above crack tips considered by
Nick et al. (2010).

We show that, for surface cracks with a fixed water level below the ice surface, we generally find that very shallow cracks are
in steady state but become unstable once they reach a stress-dependent critical value that is a small fraction of the ice thickness
(Weertman, 1980; Lai et al., 2020). Once that occurs, two scenarios are possible: first the crack may continue to grow rapidly
until it spans the entire ice thickness and calving occurs; as one would expect, this is favoured by large extensional stresses
($\tau = R_{xx}/(\rho_i gH)$) and high water tables (small water table depths $\eta = h_w/H$ below the ice surface). Alternatively, crack

growth may be arrested when crack length reaches the lower boundary of an extended region of steady states. The existence of
that region of steady states is conditional on stress $\tau$ not being too large and water table depth $\eta$ not being too small.

The subsequent evolution of such a partial crack under changes in forcing parameters is more complicated. If the water
level is fixed and stress $\tau$ is increased, the crack length will increase continuously in $\tau$ either until the crack spans the entire
ice thickness (which occurs at larger water table depths $\eta$, see phase path (B) in Figure 6), or until the range of steady states

disappears with the crack tip still at a finite distance from the base of the ice shelf, leading once more to rapid crack propagation
and calving (which occurs for smaller water table depths $\eta$, see phase path (A) in Figure 6). In either case, we can identify a





critical stress $\tau$ as a function of $\eta$ (or vice versa) at which calving occurs, as shown in Figure 9: that relationship can then, in principle, be used as a calving law.

Under more general parameter changes, calving may not be as simple: for instance, a reduction in extensional stress $\tau$
will generally leave a crack that is longer than cracks that would form anew at the new, lower extensional stress, and such an overextended crack is then more susceptible to calving if the water table subsequently rises ($\eta$ is reduced, see path (C) in Figure 6). This observation underlines that calving is generally history-dependent, and a simple calving law relating a critical extensional stress to ice thickness and water depth (e.g. Schoof et al., 2017) cannot in general be found; instead a more complicated dynamic description of crack evolution (equation (38)) becomes necessary.

Even if we accept the notion of a simple calving law like the parameterization (39), we still have to contend with the question of how to determine water table depth, which becomes key in determining the stability of the ice shelf. In dimensional terms, (39) can be written in the form

$$R_{xx} = \rho_i g H \left[ 1 - \frac{\rho_w}{\rho_i} \left( 1 - \frac{h_w}{H} \right)^3 \right], \tag{40}$$

at calving, with smaller values of extensional stress permitting the ice shelf to remain. Consider then the situation close to an
ice shelf front, where $R_{xx}$ is given by equation (25). Consider the case of a constant depth to water table $h_w$, and assume that ice thickness $H$ increases upstream of the shelf front. Suppose that the ice shelf front is at the point of calving, that is (40) is satisfied. We can ask what happens when calving occurs, in which case $H$ at the new calving front position must increase. It is however easy to show that the left-hand side of (40) with $R_{xx}$ given by (25) increases faster with $H$ than the right-hand side if the constant $h_w$ lies in the physically required range $0 < h_w < H$.

The result is that, once the critical value of $R_{xx}$ for calving is reached at the ice front, then the resulting retreat of the ice front will lead to $R_{xx}$ exceeding the critical stress given by (40), and presumably continued, rapid calving. That conclusion is however entirely dependent on the prescription of a fixed water depth $h_w$, and in reality points to the need for a more refined surface hydrology model.

As an alternative to a fixed water table, we consider the case of a fixed water volume injected into a crack. This leads
to qualitatively very different results: again there is a range of shallow steady state cracks that become destabilized once a stress-dependent critical value is reached. Unlike the case of a fixed water level, the subsequent rapid growth of the crack is generally not arrested at low extensional stresses until the crack spans the entire ice thickness. Counter-intuitively, the crack may stop growing part-way across the ice only if the extensional stress is large enough (Figure 6c): this occurs because a larger extensional stress corresponds to wider crack, and hence the same water volume corresponds to a lower water level (and
therefore water pressure).

While it may be tempting to conclude that injection of fixed water volumes invariably leads to calving if the initial crack is long enough, this may not be so: if the model were extended into the third dimension, crack propagation forced by a fixed water volume is likely not cause the crack tip to advance uniformly, but to drive a fingering instability that leads to the crack propagating all the way to the lower boundary only locally, allowing water to drain out without causing the crack
to reach the lower ice boundary everywhere and therefore without causing calving (Touvet et al., 2011; Peirce, 2016). Such





a fingering instability is driven by the larger water pressure in locations where the crack has already propagated further. A similar instability could occur where fracture propagation is driven by a fixed water level, but this is unlikely to stop the crack propagating completely where it has advanced less far as in the fixed volume case of (Touvet et al., 2011): in the fixed volume case, water level drops when the crack tip advances unevenly, potentially *reducing* the stress intensity factor in those areas

where the crack tip has not propagated as far, leaving them stranded in the sense of not advancing further; for the fixed water level case, this does not happen.

Perhaps the most important insight into the model however comes from the case of a single basal crevasse, previously considered by van der Veen (1998b) and in the extended data of Lai et al. (2020, Figure 4c). As in Lai et al. (2020), we find that basal cracks propagate across the entire ice thickness at a critical value of $\tau = R_{xx}/(\rho_i gH)$, regardless of the scaled stress

intensity factor. Computationally, we find a critical value of $R_{xx} = 0.039\rho_i gH$, while a purely theoretical argument (appendix B) puts the critical value at $R_{xx} = 0.0367\rho_i gH$. This is however a problem: the boundary condition on extensional stress at an ice shelf front is $R_{xx} = 0.05\rho_i gH$, implying that the stress there is necessarily above the critical value, and calving must occur provided small cracks are present to initiate crevasse growth.

Obviously, real ice shelves are known to persist for long periods of time, and there must be an issue with the model used

here, in van der Veen (1998a,b), and Lai et al. (2020). A closer look at the the mechanism by which calving through the growth of basal cracks occurs in the model is required. Ultimately, calving occurs as the result of torques generated on the crack faces as described in appendix B. If the net torque on the crack faces acts to open the crack, then it must be balanced by torques on the remaining neck of ice above the crack. As crack length approaches the ice thickness, the neck of ice becomes very small and torque balance requires that the stresses in the neck of ice become very large, and so too the stress intensity factor. In other

words, with a narrow neck of ice, a net torque acting to open the crack will necessarily lead to calving.

The balance of torques we have just described however results from the zero-stress boundary conditions imposed at the sides of our domain. Here, the lack of representation of buoyancy effects in our model becomes relevant. In reality, the far field in our model is a "matching region" with a buoyantly supported thin elastic beam, which can support torques and shear forces (Sayag and Worster, 2011; Wagner et al., 2016; Warburton et al., 2020). In such a thin plate, vertical displacements cause a

non-negligible imbalance between gravitational force on an ice column and the water pressure that must be balanced by a gradient in shear force; at the ice thickness scale described by our model, such buoyancy effects are absent (see also Buck and Lai, 2021).

As Figure 10 shows, solutions to the model used here typically correspond to solutions with non-zero vertical displacement gradients $\partial u_3/\partial x$ (that is, with a definite tilt) in the far field, causing the ice to emerge progressively from the water in the far

field (as shown) or be submerged progressively. The adjacent beam-like part of the shelf will therefore be subject to buoyancy effects as described above. It is realistic to expect that these, in turn, will lead to far-field stresses imposed on the domain considered in our model, which cannot be uncoupled from the elastic beam region, and these stresses are likely to generate a torque that may stabilize the crack against the torque-driven calving at low extensional stresses $R_{xx}$ that our model without far field stresses currently predicts. We leave the problem of coupling the domain considered here to a beam-like far field to future

work.





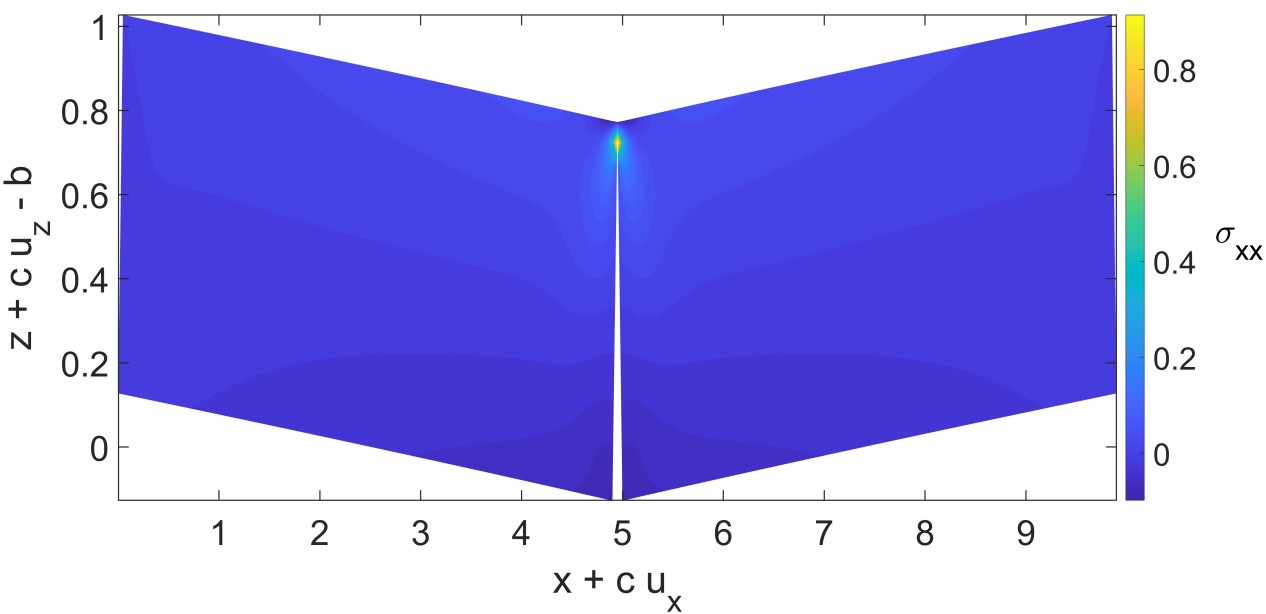

**Figure 10.** The extensional stress $\sigma_{xx}$ distribution for a basal crack with $\tau = 0.06$ and $d_b = 0.9$. We illustrate the results here using a deformed domain, plotting $\sigma_{xx}$ against a rendering of Eulerian position $(x + cu_x, z + cu_z)$, where we use $c = 0.02$ for clarity (in reality, actual displacements are smaller, with $c = \rho_i gH/E' = 0.004$ for $H = 500$ m and $E' = 1.1 \times 10^9$ Pa).

There are additional improvements to our model that need to be addressed by future work. The crack propagation rate in our model based on a quasi-static crack propagation description due to Freund (1990). The latter most likely ought to be replaced by a hydrofracture-type description of fracture propagation (Spence and Sharpe, 1985; Tsai and Rice, 2012) in future work, in which the rate of crack propagation is controlled by changes in water pressure that occur as the crack expands and water

has to flow to fill expanded crack. For the case of a single crack propagating through the ice, the distinction between the two formulations is however moot: in either case, the crack will grow if the statically computed stress intensity factor exceeds the fracture toughness of ice (Zehnder, 2012; van der Veen, 1998a,b; Lai et al., 2020).

More significantly, the computations reported in this paper rely on a particularly simple form of the viscous pre-stress $\sigma_{ij}^v$, and a prescribed, parallel-sided slab geometry into which cracks are incised. Once there is a crack across part of the ice

thickness, neither assumption will remain valid: elsatic stresses will decay relatively quickly after crack propagation (over the Maxwell time scale), leading to an adjustment in the viscous pre-stress $\sigma_{ij}^v$ to ensure continued force balance, while ice geometry will adjust more slowly over an advective time scale, comparable with the time ice takes to traverse the ice shelf. The adjustment in viscous pre-stress and in ice geometry require coupling the model for crack propagation described here with a model for viscous flow (see also Yu et al., 2017).



## Appendix A: Boundary Element Discretization

The elastic stress $\sigma_{ij}^e$ satisfies $\partial \sigma_{ij}^e / \partial x_j = 0$, and can consequently be written in terms of the Airy stress function $\Phi$ (Rice, 1968) as

$$\sigma_{xx}^e = \frac{\partial^2 \Phi}{\partial z^2}, \quad \sigma_{zz}^e = \frac{\partial^2 \Phi}{\partial x^2}, \quad \sigma_{xz}^e = -\frac{\partial^2 \Phi}{\partial x \partial z}, \tag{A1}$$

Differentiation and substitution equation into (7) gives us the biharmonic equation $\nabla^4 \Phi = 0$.

We define a dislocation at a point $(x_0, z_0)$ with orientation $\mathbf{n}$ as the limit a short crack with normal direction $\mathbf{n}$ at a location $(x_0, z_0)$, with a unit integral of displacement. We define a discontinuity in $\mathbf{u}$ such that $\mathbf{u}^+ \cdot \mathbf{n}^+ - \mathbf{u}^- \cdot \mathbf{n}^- = \delta(x - x_0)\delta(z - z_0)$, with the component of $\mathbf{u}$ perpendicular to $\mathbf{n}$ and the corresponding components of $\sigma'$ continuous at $(x_0, z_0)$. Owing to the homogeneity and isotropy of the underlying problem, we can use a translation and rotation to map the displacement discontinuity onto the origin, with the normal direction parallel to the $z$-axis. Let the relevant transformed coordinates be $(x', z')$, where $x' = (x - x_0)\cos(\theta) + (z - z_0)\sin(\theta)$ and $z' = -(x - x_0)\sin(\theta) + (z - z_0)\cos(\theta)$, $\theta$ being the angle of rotation respect to the global coordinates $(x, z)$. We make the additional assumption that displacements $u$ vanish at infinity, which resolves the non-uniqueness in the relationship between a given $\Phi$ (and therefore, strain rate $\varepsilon$) and the corresponding $u$. The imposition of displacement boundary conditions relies on integrating

$$\frac{\partial u_2'}{\partial x_2'} = \frac{\partial u_z'}{\partial z'} = \varepsilon_{22} = \frac{\partial^2 \Phi}{\partial z'^2} - \frac{\nu}{1 - \nu}\nabla^2 \Phi, \qquad \frac{\partial u_1'}{\partial x_1'} = \frac{\partial^2 \Phi}{\partial x'^2} - \frac{\nu}{1 - \nu}\nabla^2 \Phi, \tag{A2}$$

from infinity.

In the transformed coordinate system, we can solve for $\Phi$ corresponding to the displacement discontinuity at the origin formally using Fourier transforms. Defining in the usual way that $\hat{\Phi} = \int_{-\infty}^{\infty} \Phi(x', z') \exp(-ikx') dx'$, $\hat{\Phi}$ vanishing at infinity takes the form $(a_+ z' + b_+)\exp(-|k|z')$ for $z' > 0$, $(a_- z' + b_- z')\exp(|k|z')$ for $z' = 0$, with the coefficients $(a_+, b_+, a_-, b_-)$ determined by the requirements that $d^2\hat{\Phi}/dz'^2$, $-k^2\hat{\Phi}$ and $\hat{u}_x'$ are continuous while $[\hat{u}_z']_-^+ = 1$, where $[\cdot]_-^+$ simply represents the difference between limiting values take as $z' \to 0$ from above and below, and $u_x'$ and $u_z'$ are computed as described above. The solution for $\hat{\Phi}$ becomes $\hat{\Phi} = E'/[4(|k| + z')]\exp(-|k|z')$ for $z' > 0$, and $\hat{\Phi} = E'[/4|k| - z')\exp(|k|z')$ for $z' < 0$. While a closed-form solution for $\Phi$ is not available, displacements and stresses as functions of position are simple to compute from the Fourier transform solution $\hat{\Phi}$. We can repeat the same procedure to derive the solution displacement discontinuity parallel to the surface, and then transform the results back to $(x, z)$.

Assume that we can parameterize the boundary $\partial\Omega$ in terms of an arc length coordinate $s$ as $(x, z) = (X(s), Z(s))$. Consider a delta-function like displacement discontinuity some $(X(s'), Z(s'))$ on the boundary, with a normal component $D_n$ and tangential component $D_t$. We can use the Green's functions described above to compute the displacements and stresses at any part of the boundary generated by the displacement discontinuity components on another part of the boundary. Computing shear and normal stress at the boundary at $\mathbf{r} = (X(s), Z(s))$ due to a displacement discontinuity at $\mathbf{r}' = (X(s'), Z(s'))$ with normal and tangential components $(D_n, D_t)\delta(s - s')$ corresponds to computing the stress field at $x' = (\mathbf{r} - \mathbf{r}') \cdot \mathbf{t}'$, $z' = (\mathbf{r} - \mathbf{r}') \cdot \mathbf{n}'$ in our transformed coordinate system, where $\mathbf{t}'$ and $\mathbf{n}'$ are tangent and normal unit vectors at $\mathbf{r}'$, and the computing the normal and





tangential components of those stresses at $\mathbf{r}$. The displacements can be calculated as integrals over displacement discontinuities on the boundary of the form

$$u_n(s) = \oint G_n^n(s,s')D_n(s') + G_n^t(s,s')D_t(s')\mathrm{d}s', \qquad u_t(s) = \oint G_t^n(s,s')D_n(s') + G_t^t(s,s')D_t(s')\mathrm{d}s'. \tag{A3}$$

The stresses can be computed as integrals over displacements discontinuities on the boundary, of the form

$$\sigma_{nn}(s) = \oint F_{nn}^n(s,s')D_n(s') + F_{nn}^t(s,s')D_t(s')\mathrm{d}s', \qquad \sigma_{nt}(s) = \oint F_{nt}^n(s,s')D_n(s') + F_{nt}^t(s,s')D_t(s')\mathrm{d}s'. \tag{A4}$$

As indicated, the precise form of the functions $F_{nn}^n$ $F_t^t$, $F_{nt}^n$ and $F_{nt}^t$ depends on the shape of the boundary, but can be derived from the formulas for stresses at arbitrary locations $(x', z')$ due to a point dislocation at the origin in the transformed coordinate system. If the boundary is smooth at the point $(X(s), Z(s))$, then for instance the formula for $\sigma_{z'z'}'$ above (A4) implies that

$F_{nn}^n \sim E'/(4\pi)1/(s-s')^2$. The integrals in (A4) are hypersingular, and need to be understood in the sense of Hadamard (Ang, 2013).

Computationally, we approximate $\partial\Omega$ as consisting of $N$ discrete straight line segments, treating $D$ as piecewise constant on each, and using a collocation approach, forcing $\sigma_{nn}$, $\sigma_{nt}$ and $D$ to take the imposed values at the centre of the same line segments. With $D$ piecewise constant on a given line segment, we can handle the hypersingular integral as follows: take for

instance the integral

$$\int_{\Gamma_i} F_{nn}^n(s_i - s')D_n(s')\mathrm{d}s' \sim \int_{\Gamma_i} \frac{E'}{4\pi}\frac{1}{(s_i - s')^2}D_n(s')\mathrm{d}s', \tag{A5}$$

over the boundary segment $\Gamma_i$ whose center corresponds to $s = s_i$; the hypersingular integral is to be understood as $\int_{\Gamma_i} D(s')/(s-s')^2\mathrm{d}s' = -\mathrm{d}/\mathrm{d}s PV \int_{\Gamma_i} D(s')/(s-s')\mathrm{d}s$, where 'PV' indicates the usual Cauchy principal value. If we treat D as piecewise constant and and evaluate it at $s_i$, we obtain

$$\frac{E'}{4\pi}\int_{\Gamma_i} \frac{D_n(s')}{(s_i - s')^2}\mathrm{d}s' = -\frac{E'}{4\pi}\frac{D_{n,i}(s_i^+ - s_i^-)}{(s_i^+ - s_i)(s_i - s_i^-)}, \tag{A6}$$

where $s_i^+$ and $s_i^-$ are the end points of the segment $\Gamma_i$. In evaluating $\sigma_{nn}$ at $s_i$, the integrals over all other boundary elements correspond to non-singular integrals and can be computed directly. The result of the procedure is that we relate the element values $D_n^i$ and $D_t^i$ linearly to the corresponding cell centre stresses $\sigma_{nn,i}$ and $\sigma_{nt,i}$ as $\sigma_{nn,i} = \sum_j(F_{nn,ij}^n D_{n,j} + F_{nn,ij}^t D_{t,j})$ and $\sigma_{nt,i} = \sum_j(F_{nt,ij}^n D_{n,j} + F_{nt,ij}^t D_{t,j})$. We obtain $2N$ discrete equations by requiring that $\sigma_{nt,i} = 0$ everywhere, that $\sigma_{nn}^i$

take a prescribed value at any element centre on the external part of the domain boundary, and that $\min(\sigma_{nn}^i + p_f, D^i) = 0$ on elements in the cracks. The system is semi-smooth (continuous with a piecewise continuous, in fact piecewise constant, Jacobian) and we solve it using a semi-smooth Newton method, equivalent to the following procedure: every step in the iteration, we assume that a prescribed portion of $\partial\Omega$ consists of contact areas. On these, $D_n$ is prescribed, and we solve for $\sigma_{nn}$. Contact and non-contact areas are then reassigned: any part of the contact area on which $-\sigma_{nn} < p_f$ becomes a non-

contact area in the next iteration, while any part of the non-contact portion of the crack surfaces on which $D_n < 0$ becomes a





contact area. Once we have a solution, we compute $K_I$ as (Rice, 1968)

$$k_I = \sqrt{\frac{\pi}{s_{N_c}^+ - s_{N_c}^-}} \frac{D_n^{N_c}}{8}. \tag{A7}$$

where $N_c$ is the element at the crack tip, with $s_{N_c}$ being the crack tip position.

**Appendix B: Limiting form of $K_I$ for short and long cracks**

The limiting form of the stress intensity factor for short cracks, and for "long" cracks that span nearly the entire ice thickness, is important in determining whether (and when) these can be in steady state, which can be challenging to determine computationally. The behaviour of $K_I$ for crack length $d$ approaching the ice thickness is particularly relevant to the style of calving that occurs for basal cracks as well as surface cracks subject to large extensional stresses and low water levels. We consider both cases in turn, explaining how the limiting forms can be determined from simple scaling arguments. We note that these

limiting forms are explicitly built into the interpolated Green's function (Tada et al., 2000) used in Lai et al. (2020), but an explicit description is useful to contextualize our results.

In both cases, the scaling argument reduces either the crack (if short) or the remnant "neck" of ice (if the crack nearly spans the ice thickness) to a boundary layer, subject either to vanishing far field forcing for the short crack, or to a finite torque and force for the short neck of ice.

**B1   Short Cracks**

This case was previously considered in Weertman (1980), and requires little elaboration. For shallow depths and finite $\tau$, the pre-stress for a shallow crack at leading order is simply $\sigma_{ij}^v n_j \sim \tau n_i$ in dimensionless terms for a surface crack, and $(\tau - 1 + r^{-1})n_i$ in a bottom crack. Treating these as $O(1)$, all that is required to obtain the relevant boundary layer formulation is to rescale position as $(X, Z) = (x - W/2, z - s)/d$ or $(X, Z) = (x - W/2, b - z)/d$, where $d \ll 1$ is the crack length, and

displacement as $U_i = u_i/d$, while leaving the stress field unchanged as $\Sigma_{ij} = \sigma_{ij}^e$. The result, at leading order in $d$, is a half-space problem in $(X, Z)$ with dependent variables $U_i$. In the $(X, Z)$ coordinate system, the crack has unit length and a normal stress of $\Sigma_{ij} n_i n_j = -\tau$ if considering a near surface crack, or $\Sigma_{ij} n_i n_j = -\tau + 1 - r^{-1}$ for a short basal crack. The outer boundary at $Z = 0$ is traction-free and far-field stress also vanishes. Since these normal stresses are constant, linearity demands that stress is proportional to the normal stress on the crack face, and so is the rescaled stress intensity factor $\tilde{K}_I = K_I d^{-1/2}$;

moreover, with a unit crack length and the leading order boundary layer problem being independent of $d$, $\tilde{K}_I$ is independent of $d$. Consequently, $K_I = d^{1/2} \tilde{K}_I \sim d^{1/2} \tau$ for a surface crack, and $K_I \sim d^{1/2}(\tau - 1 + r^{-1})$ for a basal crack. As a result, with a non-zero $\kappa$, short cracks (of length $\lesssim \kappa^2/\tau^2$ for a surface crack, and $\lesssim \kappa^2/(\tau - 1 + r^{-1})$ for a basal crack) are always steady. This explains the narrow regions of short steady states in Figures 6 and 8.





## B2 Long Cracks

For a short remnant ice neck, we again rescale distance, now with $1 - d \ll 1$, so $(X, Z) = (x - W/2, z - b)/(1 - d)$ for a surface crack extending almost to the base of the shelf, and $(X, Z) = (x - W/2, s - z)/(1 - d)$ for a basal crack. The ice neck has to support an $O(1)$ force as well as an $O(1)$ torque. To support an $O(1)$ force, stresses on the ice neck have to scale as $(1 - d)^{-1}$, while stress variations have to scale as $(1 - d)^{-2}$ to generate an $O(1)$ torque. The stress field in the ice neck is consequently dominated by torques, and we rescale as $\Sigma_{ij} = (1 - d)^2 \sigma_{ij}^e$, $U_i = (1 - d) u_i$. The result is again a half-space problem, with a crack extending from $Z = 1$ to $Z \to \infty$. At leading order in $(1 - d)$, the crack and the outer boundary at $Z = 0$ are again traction-free, and the forcing on the problem takes the form of an applied far-field torque within the ice. By similar argument as for long cracks, we can conclude that the rescaled stress intensity factor $\tilde{K}_I = (1 - d)^{3/2} K_I$ is proportional to that applied torque, and independent of $(1 - d)$, leading to the conclusion that $K_I \sim (1 - d)^{-3/2} \times$ the applied torque.

This suggests that $K_I$ diverges to $\pm \infty$. The description of the half-space model with a crack with vanishing traction extending from $Z = 1$ to $Z = \infty$ in the last paragraph however assumes that the crack is open and subject to stress boundary conditions near its tip. In reality, negative values of $K_I$ are of course not possible if we impose the contact constraint. As a result, we actually find that either $K_I$ diverges to positive infinity if the crack is open close to the crack tip, or that $K_I$ vanishes and the crack is closed. Which of the two cases applies simply depends on the sign of the applied torque.

Assume for a moment that there are no contact areas in the crack, which is of course only self-consistent if the resulting torque leads to a positive $K_I$. In that case, the normal stresses on the crack are known and the torque $T$ on the remnant neck of ice is easy to calculate at leading order in $(-1)$. Take first the case of surface cracks with constant water levels. In that case, the torque is

$$T = 2 \int_b^s [\tau - (s - z)](z - b) \mathrm{d}z + 2 \int_b^{s-\eta} r^{-1}(s - \eta - z)(z - b) \mathrm{d}z = \tau + \frac{r^{-1}(1 - \eta)^3 - 1}{3},$$

where $s = (1 - r)$, $b = r^{-1}$, a positive value of $T$ indicating that the torque is opening the crack and therefore corresponding to $K_I \to +\infty$ as $d_t \to 1$. If calving is the result of a remnant neck of ice being fractured by the applied force, this leads to the calving law

$$\tau = \frac{1 - r^{-1}(1 - \eta)^3}{3},$$

We can compare this result to Figures 6b and 9. For an empty crack, $\eta = 1$, we got $\tau = 0.34$, numerically and from formulation above we get $\tau = 1/3 = 0.33$.

Similarly, for a basal crack that penetrates almost to the upper surface, the torque is instead

$$T = 2 \int_b^s [\tau - (s - z)](s - z) \mathrm{d}z + 2 \int_b^0 r^{-1} z(s - z) \mathrm{d}z = \tau + \frac{1 - r^{-1}(1 - (1 - r)^3)}{3},$$

suggesting a calving law

$$\tau = \frac{r^{-1}(1 - (1 - r)^3) - 1}{3} = 0.0367.$$

Again here, we can compare our result to the numerical one in Figure 8b which we get $\tau = 0.039$ for $d_b = 1$.



*Code availability.* The code for calculations are available from the corresponding author upon request.

*Author contributions.* MZ executed the research. MZ and CS designed the project and wrote the paper. MZ and AP developed the numerical method and the code. All authors edited the paper.

*Competing interests.* The authors declare no competing interests

*Acknowledgements.* CS would like to thank Eric Dunham and John Platt for advice regarding dynamic crack propagation. MZ was supported by the ArcTrain NSERC CREATE graduate training program and by NSERC Discovery Grant funding to CS. CS acknowledges NSERC Discovery Grant RGPIN-2018-04665. AP acknowledges the support of NSERC Discovery Grant RGPIN-2015-06039.





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
