# Peer review of "The effect of hydrology and crevasse wall contact on calving"

_The Cryosphere, 2022_

## Author Comment (AC1)

**The effect of hydrology and crevasse wall contact on calving**

Maryam Zarrinderakht[1], Christian Schoof[1], and Anthony Peirce[2]

[1]Department of Earth, Ocean and Atmospheric Sciences, University of British Columbia, BC, Canada
[2]Department of Mathematics, University of British Columbia, BC, Canada
**Correspondence:** M. Zarrinderakht (mzaryam@eoas.ubc.ca)

RC1: 'Comment on tc-2022-37', Anonymous Referee #1, 08 May 2022

This article presents a linear elastic fracture mechanics (LEFM) approach to estimate the penetration depth of water-filled crevasses in an ice shelf. The key novelty is that the authors consider the introduction of crevasse generates elastic stress in the ice shelf, which is otherwise at equilibrium due to viscous stress. The proposed model is an improvement over the van der Veen (1998a,b) and Lai et al. (2020) in that it considers crack wall contact using the discontinuity boundary element method. With regard to water in crevasses, the paper considers both fixed water table and fixed water volume injected, which leads to different propagation conditions. The article is generally well written from Section 3 onwards, but Section 1 and 2 have a few typos and confusing sentences, which can be easily fixed. The conclusion of the paper is long and a bit hard to follow. Overall, I found the article is a good contribution and I recommend it for publication with minor revisions.

*Our response to the referee is in red and italicized.*

Detailed Comments:

1. The introduction can be improved, as I found a few typos and grammatical errors. Also, it does not acknowledge a lot of prior work on this topic. For example, the article cites Lipovsky (2020) for numerical approaches for LEFM, but it was previously introduced in an article by Jimenez and Duddu (2018). https://www.cambridge.org/core/journals/journal-of-glaciology/article/on-the-evaluation-of-the-stress-intensity-factor-in-calving-models-using-linear-elastic-fracture-mechanics/0378315BDB37E88E37B1B07F6BC60426

   *Some more literature reviewed and cited.*

2. Replace the usage of the word "torque" with "moment". In physics, the turning effect of a force is generally termed as torque, but in mechanics torque stands for torsional moment, whereas the seawater pressure on an ice shelf causes a bending moment.

   *In our understanding, the definition of 'torque' suggested by the referee is not universal to continuum mechanics, but is indeed typical usage in continuum mechanics as practiced by mechanical engineers, see*
   *https://en.wikipedia.org/wiki/Torque*

   *Counterexamples using 'torque' in the same sense as we do within glaciology can be foundin e.g. Lipovsky, B.: Ice Shelf Rift Propagation: Stability, Three-dimensional Effects, and the Role of Marginal Weakening, The Cryosphere, 14, 1673–1683, 2020.*

*Scambos, T., Fricker, H.A., Liu, C.-C., Bohlander, J., Fastook, J., Sargent, A., Massom, R., and Wu, A.-M.: Ice shelf disintegration by plate bending and hydro-fracture: Satellite observations and model results of the 2008 Wilkins Ice Shelf break-ups, Earth Planet. Sci. Lett., 280, 51–60, 2009.*

*Evatt, G. and A.C. Fowler, A.C., Cauldron subsidence and subglacial floods, Ann. Glaciol., 45, 163–168, 2007*

*For consistency with this earlier work, we intend to keep using 'torque' as in the original submission.*

3. The model considered here is not a Maxwell model, as mentioned on page 3, line 75. In a Maxwell-type, the viscous stress must be equal to the elastic stress. The strains are additively split. I believe the assumption of this paper is a compressible Kelvin-type model. The introduction of the crack within an otherwise viscous ice shelf at equilibrium leads to elastic stress perturbations. These elastic stress vanish on the boundary far away from the crack. This is better clarified elsewhere in the paper, but not in the model description early on.

*We disagree. The model we use is the appropriate limit of an elastically compressible Maxwell model for the case of an imposed change in stress that occurs over a time scale much longer than the inertial time scale (as discussed in the paper) and much shorter than a single Maxwell time. That change in stress obviously occurs here due to the change in boundary conditions introduced by propagation of the crack. As the referee states, elastic and viscous stresses are the same and strain is additive. The point is that, if there is an abrupt change in stress field relative to a pre-existing stress, then the viscous strain is negligible at the time scale under consideration (much less than a single Maxwell time). As a result the* change in stress *can simply be related to elastic strain through an elastic rheology. Meanwhile, if the pre-existing stress (labelled the "pre-stress" in the paper) was slowly varying prior to propagation to the crack, then that pre-stress is related to the velocity field that existed just prior to the propagation of the crack by a simple viscous rheology.*

*Mathematically, an elastically compressible upper-convected Maxwell fluid rheology can be written in the form*

$$\frac{(1+\nu)\delta_{ik}\delta_{jl} - \nu\delta_{ij}\delta_{kl}}{E}\overset{\triangledown}{\sigma}_{kl} + \frac{1}{2\eta}\left(\sigma_{ij} - \frac{1}{3}\sigma_{kk}\delta_{ij}\right) = D_{ij}. \tag{1}$$

*where $\eta$ is viscosity, $v_i$ is velocity, $D_{ij} = (\partial v_i/\partial x_j + \partial v_j/\partial x_i)/2$ is strain rate, and the superscript $\triangledown$ denotes the usual upper-convected derivative*

$$\overset{\triangledown}{\sigma}_{ij} = \frac{\partial \sigma_{ij}}{\partial t} + v_k\frac{\partial \sigma_{ij}}{\partial x_k} - \frac{\partial v_i}{\partial x_k}\sigma_{kj} - \frac{\partial v_j}{\partial x_k}\sigma_{ik}, \tag{2}$$

*In the limit of a an abrupt change in stress, this translates into a large derivative $\partial \sigma_{ij}/\partial t$, so the left-hand side of (1) is dominated by the time derivative, and can be approximated by*

$$\frac{(1+\nu)\delta_{ik}\delta_{jl} - \nu\delta_{ij}\delta_{kl}}{E}\frac{\partial \sigma_{kl}}{\partial t} = D_{ij},$$

*Integrating from an initial time $t_i$ at which the crack starts propagating (so $\sigma_{ij}(x,t_i)$ is the pre-existig stress before introduction of the crack),*

$$\frac{(1+\nu)\delta_{ik}\delta_{jl} - \nu\delta_{ij}\delta_{kl}}{E}[\sigma_{ij}(x,t_f) - \sigma_{ij}(x,t_i)] = \varepsilon_{ij}(x) = \int\limits_{t_i}^{t_f} D_{ij}(x,t)\partial t.$$

*The time integral over the strain rate simply becomes the strain $\varepsilon_{ij}$ accumulated over the time period in question, and we can write the*

$$\sigma_{ij}(t,x) = \sigma_{ij}^v + \sigma_{ij}^e.$$

*where $\sigma_{ij}^v = \sigma_{ij}(t_i,x)$ is the pre-stress (related viscously to the velocity field $v_i(t_i,x)$ that was present before the introduction of the crack) and $\sigma_{ij}^e$ is an effectively elastic stress that satisfies an elastic rheology of the form*

$$\frac{(1+\nu)\delta_{ik}\delta_{jl} - \nu\delta_{ij}\delta_{kl}}{E}\sigma_{ij}^e = \varepsilon_{ij}.$$

*This is consistent with the formulation used in the paper. We have added additional text to this effect to the paper.*

4. Line 98, page 4, it is mentioned the stress field defined by (6) cannot be generated by an elastic rheology is not true. This stress field can be obtained with a nearly incompressible elastic rheology. It is really not the elastic or viscous nature but rather the incompressibility assumption that leads to this stress state. Please see Sun et al. (2021) Appendix A for the derivation of the elastic stress field, wherein if you plug in Poisson's ratio of 0.5, you would recover the stress field defined by (6). https://www.sciencedirect.com/science/article/abs/pii/S2352431621000626

   *This is true if you assume that $\nu = 0.5$ is viable, and the medium in question is elastically incompressible. We have altered the text to state that the assumed far field stress is incompatible with an elastic rheology unless ice is assumed to be elastially incompressible with a Poisson's ratio of 0.5. Empirically determined values of $\nu$ for ice are however around 0.3.*

5. Line 119 - 120, page 5, seems like a typo, there is no subscript on $[v]_-^+$ and in equation (9) u should not be bold in $[u]_-^+$.
   *That is true, we have corrected this expression.*

6. (11) seems to have some wrong notation. The index j appears three times and this violates Einstein's summation convention.
   *We have changed this equation to the correct form of $(\delta_{ij} - n_i n_j)\sigma_{jk}n_k = 0$.*

7. Line 139, page 5, I do not understand why it is more natural to prescribe water volume. Isn't it as poorly constrained as the water height in crevasses. Please explain how one would constrain water volume from observations.
   *In our view of the dynamics of crevasse propagation, the relevant question is not how one constraints the forcing parameters observationally, though that is an entirely relevant question once you have a successful, self-consistent theory. The relevant question at the the level of model development in our view is how water level is controlled physically. In particular, while a crevasse propagates, we need to know whether water level in the crevasse remains constant, or if it changes, what controls water level. The two cases we consider are essentially end-members hydrologies: A fixed water level $h_w$ represents a large water reservoir (such as a porous subsurface aquifer) that easily drains and buffers water level in crevasses against changes. A fixed water volume is what we would expect in a situation where there is no water reservoir and no significant water supply to the crevasse on the time scale of propagation — presumably, on time scales less than a single Maxwell time. We will amend the text of the paper to make this clearer. Reality is, of course, likely to*

*lie somewhere between the two, but requires a more refined understanding of near-surface hydrology than we are aiming*

80 *for here.*

8. Line 159, page 6, please use text roman i for the subscript for ice density, so that it does not mix up with the subscript index i.

   *Done.*

9. Line 205, page 8, The authors state that instead of solving a full dynamic crack problem, they can use the semi-analytical

85 theory of Freund (1990). It is not clear to me why Freund's approach is needed. Please explain why the simple stability criterion for steady state crack used in Lai et al. (2020) is not adequate for analysis.

   *It is correct that, for a single crack, a simple stability criterion of "the crack propagates if $K_I > K_{I,c}$" is satisfied. This is acknowledged on line 187 onwards of the original submission: "If there is only a single crack, its dynamics under changes in parameters are likely to be simple: any such change that reduces the static $K_I$ below $K_{Ic}$ would*

90 *leave the crack length unchanged, while increases in static $K_I$ above $K_{Ic}$ would cause lengthening of the crack until its tip once more attains the static value of $K_{Ic}$." The formulation here is developed specifically with a view to being able to model multiple competing cracks (as alluded to in line 192 onwards of the original submission, "In the case of multiple cracks, however, not all cracks need to propagate simultaneously or at the same speed, and it is then unclear which cracks should be lengthened until they reach $K_{Ic}$"), We intend to treat that case in a follow-up paper currently in*

95 *preparation. Once there are competing cracks that might lengthen simultaneously, it becomes important to know how fast they propagate relative to each other, (Put another way, in a one-dimensional dynamical system, all I need in order to be able to understand the qualitative, long-term behaviour of the system is (apart from some constraints on smoothness) to know when the right-hand side is positive, negative, and zero. That is no longer true in higher dimensions.)*

   *In order not to have to repeat the basic statement of the model and nondimensionalization in detail, we would like to*

100 *keep the formulation as is, allowing for that generalization to be done more efficiently.*

10. In Eq. (22), the quantity $[t]$ comes out to be negative. Is that correct?

    *No. $[t]$ is always positive since $K'(0)$ is negative. $K'(0)$ is the second term in the Taylor expanding of the universal function $K(\dot{d})$. Please see the Universal function, $K(\dot{d})$, versus crack tip velocity in Chapter 6, Freund (1990).*

11. In Eq. (32) you have the term $(s - \eta \breve{} z)$ where z is has a dimension but $\eta$ is nondimensionalized. Is there any typo there?

105 *In line 248 we state that from now on we are omitting the asterisks from the nondimensionalized variables. Both s and z are nondimensionalized in this equation.*

12. I found the results section to be a bit hard to read. I felt like a lot of minor details were discussed which at time made me lose the big picture. I think the paper can be condensed a lot in this section.

    *We suspect our goal differs somewhat from what the referee has in mind. Given that the basic idea behind the model*

110 *was previously developed in van der Veen's (1998) papers as well as Lai et al (2020), our intention was not primarily a "big-picture" article but a detailed and comprehensive analysis. We trust that such an approach still has its place in the*

*field and in the pages of* The Cryosphere, *even if it places greater demands on the reader (who we assume will typically be someone more deeply invested in the study of calving).*

13. Throughout the paper, I found minor typographical errors that are a few too many, but I did not want to list them here. Please proofread the entire article before submitting the final version.

    *Our apologies. We have done our best to re-proofread the paper and eliminate more (if possibly not all!) of the remaining typos.*

14. In section 5, two calving laws were introduced one for basal crevasses and another for surface crevasses. A major critique is that unless these calving laws are incorporated in an ice sheet model and validated with observational data, we do not know if it is good. However, this maybe beyond the scope of this article.

    *Indeed, our goal for the time being was to construct and — more significantly — analyze a prognostic model for calving based on simple but self-consistent physical ingredients that are compatible with incorporation into a large-scale ice sheet model. Calibration, validation and testing are beyond the scope of the present work, but would be an important step.*

15. The conclusion of this paper is really long and I found it difficult to read. It will be good if it can be broken up into subsections to improve readability.

    *We made some changes and some subsections are added. The intention here was to be comprehensive.*

---

## Author Response (AR1)

**The effect of hydrology and crevasse wall contact on calving**

Maryam Zarrinderakht[1], Christian Schoof[1], and Anthony Peirce[2]

[1]Department of Earth, Ocean and Atmospheric Sciences, University of British Columbia, BC, Canada
[2]Department of Mathematics, University of British Columbia, BC, Canada

**Correspondence:** M. Zarrinderakht (mzaryam@eoas.ubc.ca)

RC1: 'Comment on tc-2022-37', Anonymous Referee #1, 08 May 2022

This article presents a linear elastic fracture mechanics (LEFM) approach to estimate the penetration depth of water-filled crevasses in an ice shelf. The key novelty is that the authors consider the introduction of crevasse generates elastic stress in the ice shelf, which is otherwise at equilibrium due to viscous stress. The proposed model is an improvement over the van der Veen (1998a,b) and Lai et al. (2020) in that it considers crack wall contact using the discontinuity boundary element method. With regard to water in crevasses, the paper considers both fixed water table and fixed water volume injected, which leads to different propagation conditions. The article is generally well written from Section 3 onwards, but Section 1 and 2 have a few typos and confusing sentences, which can be easily fixed. The conclusion of the paper is long and a bit hard to follow. Overall, I found the article is a good contribution and I recommend it for publication with minor revisions.

Our responses to the referee are in red, with quotations from the manuscript in italics.

Detailed Comments:

1. The introduction can be improved, as I found a few typos and grammatical errors. Also, it does not acknowledge a lot of prior work on this topic. For example, the article cites Lipovsky (2020) for numerical approaches for LEFM, but it was previously introduced in an article by Jimenez and Duddu (2018). https://www.cambridge.org/core/journals/journal-of-glaciology/article/on-the-evaluation-of-the-stress-intensity-factor-in-calving-models-using-linear-elastic-fracture-mechanics/0378315BDB37E88E37B1B07F6BC60426

   Some more literature reviewed and cited. We reference in the second paragraph of the introduction section of the revised manuscript:

   *A variety of different approaches have been used to model fracture in ice. Aside from early heuristic "zero-stress" type models (Nye, 1957; Nick et al., 2010; Todd and Christoffersen, 2014), these are primarily discrete element models (which do not pretend to represent ice as a continuum), linear elastic fracture mechanics models, which focus on one or a few discrete cracks, and continuum damage mechanics models, which treat calving as the result of the density of microfractures accumulating to generate a macroscopic crevasse that penetrates through the ice thickness (Larour and Aubry, 2004; Benn et al., 2007; Cook et al., 2014; Levermann et al., 2012; Borstad et al., 2012, 2013; Krug et al., 2014; Bassis and Jacobs, 2013; Mobasher et al., 2016; Yu et al., 2017; Benn et al., 2017; Todd et al., 2018). In addition to different assumptions about the basic physics involved, there are additionally different numerical approaches that can be applied*

*to the resulting models, especially in case of linear elastic fracture mechanics (Touvet et al., 2011; Tsai and Rice, 2012; Jiméneza et al., 2017; Lipovsky, 2020).*

2. Replace the usage of the word "torque" with "moment". In physics, the turning effect of a force is generally termed as torque, but in mechanics torque stands for torsional moment, whereas the seawater pressure on an ice shelf causes a bending moment.

   In our understanding, the definition of 'torque' suggested by the referee is not universal to continuum mechanics, but is indeed typical usage in continuum mechanics as practiced by mechanical engineers, see

   *https://en.wikipedia.org/wiki/Torque*

   Counterexamples using 'torque' in the same sense as we do within glaciology can be foundin e.g. *Lipovsky, B.: Ice Shelf Rift Propagation: Stability, Three-dimensional Effects, and the Role of Marginal Weakening, The Cryosphere, 14, 1673–1683, 2020.*

   *Scambos, T., Fricker, H.A., Liu, C.-C., Bohlander, J., Fastook, J., Sargent, A., Massom, R., and Wu, A.-M.: Ice shelf disintegration by plate bending and hydro-fracture: Satellite observations and model results of the 2008 Wilkins Ice Shelf break-ups, Earth Planet. Sci. Lett., 280, 51–60, 2009.*

   *Evatt, G. and A.C. Fowler, A.C., Cauldron subsidence and subglacial floods, Ann. Glaciol., 45, 163–168, 2007*

   For consistency with this earlier work, we intend to keep using 'torque' as in the original submission.

3. The model considered here is not a Maxwell model, as mentioned on page 3, line 75. In a Maxwell-type, the viscous stress must be equal to the elastic stress. The strains are additively split. I believe the assumption of this paper is a compressible Kelvin-type model. The introduction of the crack within an otherwise viscous ice shelf at equilibrium leads to elastic stress perturbations. These elastic stress vanish on the boundary far away from the crack. This is better clarified elsewhere in the paper, but not in the model description early on.

   We disagree. The model we use is the appropriate limit of an elastically compressible Maxwell model for the case of an imposed change in stress that occurs over a time scale much, longer than the inertial time scale (as discussed in the paper) and much shorter than a single Maxwell time. We give an explanation in terms of continuum mechanics below, but if you are thinking of this in terms of springs and dashpots, the point here is the following: consider a spring and dashpot in series, initially subject to a slowly varying stress (slowly compared with the Maxwell time). The strain rate in the dashpot will be given through the usual purely viscous relationship by that stress, which is the pre-stress we refer to. Suppose that there is a rapid change in stress (rapid compared with the Maxwell time). The stress *change* will be related to the additional strain in the spring by Hooke's law. It is that purely elastic relationship between strain (relative to the configuration immediately prior to the introduction of the stress change) and the stress change (which we refer to as the elastic stress) that we exploit in the paper. The pre-stress and stress change are simply added to produce the total stress. We have made two changes to the paper to give our reasoning: i) we have moved the specification of rheology forward in the model description, and i) we reference a new appendix A immediately after stating the decomposition (equation (2) in the revised manuscript:

*A derivation of this decomposition from first principles is included for completeness in appendix A*

The full text of appendix A is as follows

*Assume that ice can be treated as an elastically compressible, upper-convected Maxwell fluid, with a rheology of the form*

$$\frac{(1+\nu)\delta_{ik}\delta_{jl} - \nu\delta_{ij}\delta_{kl}}{E} \overset{\triangledown}{\sigma}_{kl} + \frac{1}{2\eta}\left(\sigma_{ij} - \frac{1}{3}\sigma_{kk}\delta_{ij}\right) = D_{ij}, \tag{1}$$

*where $\eta$ is viscosity, $v_i$ is velocity,*

$$D_{ij} = (\partial v_i/\partial x_j + \partial v_j/\partial x_i)/2, \tag{2}$$

*is the usual strain rate, and the superscript $\triangledown$ denotes the upper-convected derivative*

$$\overset{\triangledown}{\sigma}_{ij} = \frac{\partial \sigma_{ij}}{\partial t} + v_k\frac{\partial \sigma_{ij}}{\partial x_k} - \frac{\partial v_i}{\partial x_k}\sigma_{kj} - \frac{\partial v_j}{\partial x_k}\sigma_{ik}. \tag{3}$$

*Consider an abrupt change in stress due to introduction of a crack over a time scale much shorter than the Maxwell time $\eta/E$. Assuming that strains that occur over this time scale remain small, an abrupt change ins tress translates into a large derivative $\partial\sigma_{ij}/\partial t$, with the left-hand side of equation (1) dominated by the time derivative. Equation (1) can then be approximated by*

$$\frac{(1+\nu)\delta_{ik}\delta_{jl} - \nu\delta_{ij}\delta_{kl}}{E}\frac{\partial\sigma_{kl}}{\partial t} = D_{ij}.$$

*Integrating from an initial time $t_i$ at which the crack starts propagating (so $\sigma_{ij}(x,t_i)$ is the pre-existig stress before introduction of the crack),*

$$\frac{(1+\nu)\delta_{ik}\delta_{jl} - \nu\delta_{ij}\delta_{kl}}{E}[\sigma_{ij}(x,t_f) - \sigma_{ij}(x,t_i)] = \int\limits_{t_i}^{t_f} D_{ij}(x,t)dt.$$

*If the strain accumulated over the interval $(t_i, t_f)$ is small, then the integral $\int_{t_i}^{t_f} D_{ij}(x,t)dt$ at fixed $x$ is approximately the displacement of a Lagrangian particle at initial position $x$. In that case, the time integral over the strain rate simply becomes the linearized strain*

$$\varepsilon_{ij} = \int\limits_{t_i}^{t_f} D_{ij}(x,t)dt,$$

*accumulated over the time period in question. Hence we can write the*

$$\sigma_{ij} = \sigma_{ij}^v + \sigma_{ij}^e,$$

*where $\sigma_{ij}^v = \sigma_{ij}(x,t_i)$ is the pre-stress (related viscously to the velocity field $v_i(x,t_i)$ that was present before the introduction of the crack, assuming a slowly varying stress field prior to crack propagation) and $\sigma_{ij}^e$ is an effectively elastic stress that satisfies equation (4) once the plain-strain assumption is made and subscripts are restricted to run over $\{1,2\}$.*

75    4. Line 98, page 4, it is mentioned the stress field defined by (6) cannot be generated by an elastic rheology is not true. This stress field can be obtained with a nearly incompressible elastic rheology. It is really not the elastic or viscous nature but rather the incompressibility assumption that leads to this stress state. Please see Sun et al. (2021) Appendix A for the derivation of the elastic stress field, wherein if you plug in Poisson's ratio of 0.5, you would recover the stress field defined by (6). https://www.sciencedirect.com/science/article/abs/pii/S2352431621000626

80    *This is true* if *you assume that $\nu = 0.5$ is viable, and the medium in question is elastically incompressible. We have altered the text to state that the assumed far field stress is incompatible with an elastic rheology unless ice is assumed to be elastially incompressible with a Poisson's ratio of 0.5. Empirically determined values of $\nu$ for ice are however around 0.3. We have changed the wording of the paper to reflect this as follows (immediately after equation (8)):* Importantly, the stress field defined by equation (8), which on its own satisfies lateral stress conditions (6), cannot be generated by a

85    *compressible elastic rheology (in the sense that there is no displacement field that generates $\sigma_{ij}^{v}$ through equations (3)– (4) above, unless we assume that $\nu = 1/2$, corresponding to elastically incompressible ice). This explains our insistence on separation of $\sigma_{ij}$ into viscous and elastic parts $\sigma_{ij}^{v}$ and $\sigma_{ij}^{e}$, respectively.*

    5. Line 119 - 120, page 5, seems like a typo, there is no subscript on $[v]_{-}^{+}$ and in equation (9) u should not be bold in $[u]_{-}^{+}$. *That is true, though our intention had been to define $[\cdot]$ for vector valued quantities, and the revised manuscript now*

90    *defines $[\mathbf{v}]_{-}^{+} = \mathbf{v}^{+} \cdot \mathbf{n}^{+} - \mathbf{v}^{-} \cdot \mathbf{n}^{-}$ (immediately before equation (9)).*

    6. (11) seems to have some wrong notation. The index j appears three times and this violates Einstein's summation convention.
    *We have changed this equation to the correct form of $(\delta_{ij} - n_i n_j)\sigma_{jk} n_k = 0$.*

    7. Line 139, page 5, I do not understand why it is more natural to prescribe water volume. Isn't it as poorly constrained as

95    the water height in crevasses. Please explain how one would constrain water volume from observations.
    *In our view of the* dynamics *of crevasse propagation, the relevant question is not how one constraints the forcing parameters* observationally, *though that is an entirely relevant question once you have a successful, self-consistent theory. The relevant question at the the level of model development in our view is how water level is controlled* physically. *In particular, while a crevasse propagates, we need to know whether water level in the crevasse remains constant, or if it*

100    *changes, what controls water level. We have amended the text to state, in the paragraph before equation (14),*
    *We consider two basic scenarios as possible end-members of surface hydrological systems. The first is a prescribed water level $h_w$ below the ice surface as previously used by van der Veen (1998a). This is the 'wet' end member of surface hydrological systems, where a large and well-connected reservoir of water (presumably a subsurface aquifer) is able to rapidly supply water to the crack and maintain water level during crack propagation. The second scenario involves a*

105    *prescribed water volume in the top crack, representing a water-limited system with an isolated surface crack that is not rapidly resupplied with water as it lengthens; in this case, water level $h_w$ will generally drop as the crack propagates. The latter hydrology is also motivated by Nick et al. (2010), who use the somewhat more difficult-to-justify assumption of a prescribed water column height above the bottom the crack (see also Schoof et al., 2017): one would not generally ex-*

*pect the column height to be prescribed in nature, while the similar but not identical assumption of a fixed water volume simply reflects conservation of mass in an isolated surface crack. Further refinement to the two end-member scenarios is likely to be a target for future work: as we will see below, we obtain very different dynamical behaviour depending on whether water level or water volume is prescribed. Naturally, this also raises the question of how one would observationally constrain hydrology around surface crevasses. In the spirit of a forward modelling study, we focus here on the basic dynamics of the system under the stated assumptions, and leave open the question of observational validation.*

8. Line 159, page 6, please use text roman i for the subscript for ice density, so that it does not mix up with the subscript index i.

   Done; ditto for $\rho_w$.

9. Line 205, page 8, The authors state that instead of solving a full dynamic crack problem, they can use the semi-analytical theory of Freund (1990). It is not clear to me why Freund's approach is needed. Please explain why the simple stability criterion for steady state crack used in Lai et al. (2020) is not adequate for analysis.

   It is correct that, for a single crack, a simple stability criterion of "the crack propagates if $K_I > K_{I,c}$" is satisfied. This is acknowledged on line 187 onwards of the original submission: "*If there is only a single crack, its dynamics under changes in parameters are likely to be simple: any such change that reduces the static $K_I$ below $K_{Ic}$ would leave the crack length unchanged, while increases in static $K_I$ above $K_{Ic}$ would cause lengthening of the crack until its tip once more attains the static value of $K_{Ic}$ (or the crack propagates all the way through the ice).*" The formulation here is developed specifically with a view to being able to model multiple competing cracks (as alluded to in line 192 onwards of the original submission, "*In the case of multiple cracks, however, not all cracks need to propagate simultaneously or at the same speed, and it is then unclear which cracks should be lengthened until they reach $K_{Ic}$, and which cracks will have stress intensity factors below $K_{Ic}$ when a new equilibrium is reached.*"), We intend to treat that case in a follow-up paper currently in preparation. Once there are competing cracks that might lengthen simultaneously, it becomes important to know how fast they propagate relative to each other, (Put another way, in a one-dimensional dynamical system, all I need in order to be able to understand the qualitative, long-term behaviour of the system is (apart from some constraints on smoothness) to know when the right-hand side is positive, negative, and zero. That is no longer true in higher dimensions.)

   In order not to have to repeat the basic statement of the model and nondimensionalization in detail, we would like to keep the formulation as is, allowing for that generalization to be done more efficiently.

   We have amended the manuscript text by adding the following (line 217 onwards):

   *While we ultimately solve the problem only for a single crack in the present paper, the numerical method we use can deal with multiple competing cracks. For the sake of completeness, we therefore describe the method we use to capture dynamic crack propagation below, even though results for competing cracks will be presented in separate papers.*

10. In Eq. (22), the quantity $[t]$ comes out to be negative. Is that correct?

No. $[t]$ is always positive since $K'(0)$ is negative; the original submission incorrectly stated that $K'(0) > 0$ immediately after equation (19), which we have now corrected.

11. In Eq. (32) you have the term $(s - \eta^{\smile} z)$ where z is has a dimension but $\eta$ is nondimensionalized. Is there any typo there?

In line 248 we state that from now on we are omitting the asterisks from the nondimensionalized variables, so that both $s$ and $z$ are dimensionless in this equation.

12. I found the results section to be a bit hard to read. I felt like a lot of minor details were discussed which at time made me lose the big picture. I think the paper can be condensed a lot in this section.

We suspect our goal differs somewhat from what the referee has in mind. Given that the basic idea behind the model was previously developed in van der Veen's (1998) papers as well as Lai et al (2020), our intention was not primarily a "big-picture" article but a detailed and comprehensive analysis. We trust that such an approach still has its place in the field and in the pages of *The Cryosphere*, even if it places greater demands on the reader (who we assume will typically be someone more deeply invested in the study of calving).

13. Throughout the paper, I found minor typographical errors that are a few too many, but I did not want to list them here. Please proofread the entire article before submitting the final version.

Our apologies. We have done our best to re-proofread the paper and eliminate more (if possibly not all!) of the remaining typos.

14. In section 5, two calving laws were introduced one for basal crevasses and another for surface crevasses. A major critique is that unless these calving laws are incorporated in an ice sheet model and validated with observational data, we do not know if it is good. However, this maybe beyond the scope of this article.

Indeed, our goal for the time being was to construct and — more significantly — analyze a prognostic model for calving based on simple but self-consistent physical ingredients that are compatible with incorporation into a large-scale ice sheet model. Calibration, validation and testing are beyond the scope of the present work, but would be an important step. We allude to this in the new text immediately before equation (14),

*Naturally, this also raises the question of how one would observationally constrain hydrology around surface crevasses. In the spirit of a forward modelling study, we focus here on the basic dynamics of the system under the stated assumptions, and leave open the question of observational validation.*

15. The conclusion of this paper is really long and I found it difficult to read. It will be good if it can be broken up into subsections to improve readability.

We have introduced subsections on surface crevasses with fixed water level, surface crevasses wiht fixed water volume, and basal crevasses, to make this more readable. We hope this will not fall foul of the Cryosphere style guide.

**The effect of hydrology and crevasse wall contact on calving**

Maryam Zarrinderakht[1], Christian Schoof[1], and Anthony Peirce[2]

[1]Department of Earth, Ocean and Atmospheric Sciences, University of British Columbia, BC, Canada
[2]Department of Mathematics, University of British Columbia, BC, Canada

**Correspondence:** M. Zarrinderakht (mzaryam@eoas.ubc.ca)

RC2: 'Comment on tc-2022-37', Bradley Lipovsky, 18 May 2022

Dear Editorial Staff and Authors,

This manuscript by Zarrinderakht and co-authors was simply wonderful to read. It constitutes a significant advance in the
5  field of glacier fracture mechanics and is obviously a stepping stone towards bigger things. I enthusiastically support the
publication of this manuscript in the Cryosphere. I do have a few questions and comments that I hope will improve the quality
of the manuscript. Many of these draw connections to my own work on glacier fracture mechanics, which isn't to suggest
that my work be given any special pedestal, but is rather just to share how I think about some of the physics of these kind of
problems. Please feel free to take or leave this work as you see fit.

10  All the best,

Brad

Our responses to the referee are in red, with quotations from the manuscript in italics.

Questions and Commetns

1. Why assume the crack propagates slowly (i.e., equation 19)? We know very well that crevasses in ice shelves are seis-
15    mogenic. See, for example, Aster et al., 2021, who interpreted the unique seismic characteristics of certain impulse ice
shelf seismic observations to be caused by crevasses growth. In order for crevasses to generate seismic waves, they must
propagate at inertial (or near-inertial) velocities. Furthermore, the full inertial treatment of crevasse growth maintains
the form of Equation 18, it just changes the last multiplicative term on the right hand side. This situation was treated
by Lipovsky (2018) which to my knowledge is the first-, and prior to the present manuscript the only, study to examine
20    the dynamics of glacier fracture growth (albeit with horizontal propagation, although the authors will appreciate that the
math is the same). If the crack does move suddenly then water compressibility may be important (also, see below). The
equations necessary to treat compressible pressure gradient flow along hydraulic fractures were given by Lipovsky and
Dunham (2015) with application to hydraulic fractures in glaciers.

We agree that a truly complete model of calving should indeed incorporate a full dynamic treatment of crack propaga-
25    tion, with inertial terms retained in the momentum balance equations, and accounting for elastic waves. The text on lines
202 onwards of the original submission was intended to make that clear ("In a general, the computation of $K_I$ during
fracture propagation then requires a dynamic model in which inertial terms are not omitted in equation (2).") The point

is however that doing so renders the approach taken in our paper entirely inapplicable: in particular, it does not appear to us as though a simple solution for $K_I$ in terms of a small number of forcing parameters, with crack length(s) as the only dynamic variable(s), would be available: instead, we require a dynamic solution that resolves the displacement field throughout the domain and in time.

Unfortunately, the Lipovsky (2018) paper does not seem like a viable template for including inertial effects more completely in what we are attempting to do, though we may be mistaken. The way we read the latter paper suggests that wave propagation is computed using a Fourier transform in time and space, thereby avoiding any coupling with a crack whose size changes over time ("The entire geometry is assumed to be translationally invariant in the $x$-direction", appendix A1 of Lipovsky 2018). In the same vein, the analytical form of eq (14) in Lipovsky suggests the work is restricted to a nascent crack whose length is much smaller than the size of the ice mass, in that case its horizontal extent.

Our interest here is in a crack that propagates through a significant fraction of the ice thickness, and ultimately all the way across. As a result, if crack propagation does occur at seismic time scales, there can be no decoupling between crack propagation and wave propagation in an evolving domain geometry, precluding the use of Fourier transforms in time in any meaningful way (since the spatial domain changes over time!). As described above, it seems to us like a simple model with only crack length as the sole, scalar dynamic variable of time goes out of the window: instead, one would have to account for displacement everywhere in the ice as a dynamic variable of time *and* position throughout the calculation. At a single stroke, we have to go from a one-dimensonal dynamical system (or an $n$-dimensional one if modelling $n$ interacting cracks with prescribed orientation) to an infinite-dimensional one. We have amended the manuscript to say (line 229 onwards)

*Solving a time-dependent problem that captures elastic waves renders our just-stated objective of computing fracture propagation for many forcing parameters intractable, as it increases the number of dynamical degrees of freedom from the number of cracks to a dynamic displacement field throughout the domain.*

In the spirit of trying to do *something* tractable that stays as close to the underlying physics as we can afford computationally, we have opted to stick with the pseudo-static stress field approximation as described in section 2.2. We hope that the remaining text of the altered paragraph before equation (18) (" Short of solving a full dynamic crack propagation problem, we can use the semi-analytical theory of Freund (1990) . . .") would be sufficient to make our intent (and its limitations!) clear, but we will strengthen the description to make clear that the approximations we make are expedient rather than necessarily accurate.

2. I realized when reading the caption of Figure 3 (. . . even where the crack is close. . ." [sic]) that the authors assume hydrostatic pressure for what appear to be closed-off water blobs. Could the pressure be cryostatic? If so, this would provide additional reason to treat the compressibility of the water.

Closed-off water blobs can be treated in two ways. First, we can assume that roughness in the crack surface leads to a hydraulic connection being maintained with the surface, and water pressure being prescribed by the surface drainage system; this is the assumption we make here. The second way is to treat them as hydraulically isolated water bodies.

*In that case, it is natural to prescribe not a pressure as such, but a* hydrostatic *pressure distribution within the blob (so there is no pressure gradient and no water flow inside the blob) with the mean pressure to be determined. That mean pressure is then determined by the need to maintain a fixed water volume in the blob. That, however, is our second hydraulic scenario, and it requires a prescription of the size of that water volume, which cannot be determined from a water table height. To avoid adding additional, and poorly-constrained physics of the model, we chose to assume a hydraulic connection is maintained in the context of the first hydrology model involving a prescribed water table. We have amended the manuscript text to say (line 397)*

*That, however, is only one possible explanation, and isolated deeper water pockets (separated from the water table by a contact area) can conceivably behave as fixed water volumes instead, corresponding to the case of fixed $\beta$ we consider below.*

3. Section 2.1 Model description. Some readers might be interested to know that Lipovsky (2020) also used viscous pre-stresses in LEFM calculations. To my knowledge, this publication introduced these concepts in glaciological research. I gave a different physical explanation of the viscous pre-stresses but the form was mathematically identical to that used in the present manuscript. I do prefer the physical explanation given in the present manuscript, but I'm at least encouraged that the math is the same since I grappled with this for a while.

We now reference Lipovsky (2020) in this context.

4. Experience hiking around glaciers with water-filled crevasses tells us that crevasses are often up to a meter wide (or more). It is unlikely that this meter of opening is due entirely to elastic stresses, as one can calculate that this would require enormous and unrealistic stresses. The explanation for the opening is instead that the ice surrounding the crevasse has deformed through flow. The crack would have non-zero width in the absence of the elastic tensions. In this case, not all crack closure would result in contact. It is therefore worth noting that —in at least some cases— negative crack opening (i.e., crack closure) does not result in contact, and instead simply results in the crack getting narrower but not having walls that touch. I'm curious now: can the numerical method in this manuscript handle nonzero initial crevasse widths?

Yes, boundary element method is capable that and in general working with any geometry of the crack. As pointed out, these (older) water filled crevasses have likely attained their shape in part due to viscous deformation of ice after their formation. Part of our ongoing work deals with a coupled model for long-term viscous deformation and (presumably episodic) linear elastic crack propagation, starting with a viscously deformed geometry. The complications involved are however beyond the scope of this first paper.

5. If, on the other hand, the crevasse is assumed to be so narrow that the walls could touch, then fluid viscosity should become important [see again LD15]. Maybe these points are already acknowledged in line Line 150, where the reader is cautioned that more complexity in the fluid flow is warranted.

Yes, our intention there was indeed to acknowledge that a more complete hydrofracture model would be a desirable improvement over our work, with propagation speeds that are controlled at least partly by pressure gradients associated

*with the need to fill the opening crack with fluid. We would love to tackle this in future but the results of the present study still seemed worth reporting as they are. The current text says (line 166 onwards) [We] assume that negligible hydraulic potential gradients are required to drive fluid flow along the cracks as their tips move and the cracks open or close, so that water pressure can be treated as hydrostatic in each crack. The propagation criteria in the next section are built around this assumption, which gives a particularly simple way of handling the stability of cracks but likely needs to be superseded by a more sophisticated treatment of water movement in the cracks in future work (Spence and Sharpe, 1985). and we reiterate this in the conclusion section (lines 734 onwards) There are additional improvements to our model that need to be addressed by future work. The crack propagation rate in our model based on a quasi-static crack propagation description due to Freund (1990). The latter most likely ought to be replaced by a hydrofracture-type description of fracture propagation (Spence and Sharpe, 1985, Tsai and Rice, 2012) in future work, in which the rate of crack propagation is controlled by changes in water pressure that occur as the crack expands and water has to flow to fill the expanded crack.*

6. The first sentence of this section seems to imply that the width of the domain is an important parameter in the problem. I don't understand why this would be the case if Rxx is (conceptually at least) treated as a boundary condition at great distances (i.e., +/- infinity). Numerically, shouldn't the simulations be run for a sufficiently large domain width so that the solutions do not depend on this parameter?

   The intention is indeed to regard $W$ as wide, emulating an infinite strip of ice. Since actual computations require a finite value of $W$, it seemed wise to report it for the sake of reproducability. We did test whether our results were sensitive to the choice of $W$ and made sure to pick a value for which they were not. We have reworded the start of section 4.1 as

   *In this paper, we consider a single crack of length $d_\mathrm{t}$ (for a surface crack) or $d_\mathrm{b}$ (for a bottom crack) incised vertically at the midpoint of the domain $x = W/2$, for a parallel-sided slab domain of unit thickness in dimensionless terms with a wide domain width (where we have used $W = 10$ in the numerical solutions).*

7. I am confused by the results in Figure 3. The model seems to be treating the case with constant water volume, but yet the water volume clearly changes from figure 3b1 to 3b2. I think this is supposed to mean that some water is stored at the surface. But if water is stored at the surface, then the appropriate tractions ought to be applied at the surface of the glacier. Instead, the surface of the glacier is taken to be traction free (Equation 3). It seems like a rigorous treatment of this situation must either include the surface load or else omit crevasse depths that are too shallow to hold the prescribed water volume.

   Excess water is indeed assumed to be stored at the surface as stated in line 145 of the original submission ("Otherwise, if the proscribed volume cannot be accommodated in the crack, then $h_w = 0$ and the excess is stored at the surface"). The point is that, for a model with small displacements, the water volumes that can be stored in a crack (with an $O(1)$ scaled water volume value $\beta$ necessarily correspond to a very thin layer of water when spread out over the ice surface, at least if domain width $W$ is comparable to or larger than ice thickness as we do. The equivalent water layer thickness at the ice surface is then comparable to the width of the crack, which is much smaller than the ice thickness. The hydrostatic

130  water pressure generated by this layer is small compared with the water pressure generated by a vertical water column in any crack whose length is much greater than its width. As a result, we ignore the effect of the surface water volume on stress boundary conditions. We have added the following text (line 162 onwards)

*In putting $h_{\mathrm{w}} = 0$ when the prescribed volume cannot be accommodated in the crack, we assume that $V_{\mathrm{w}}$ remains comparable to the volume that can be stored in a crack, and therefore corresponds to an insignificant water depth at the*
135  *ice surface when ponded in this way. "Insignificant" here implies that ponded water depth is small compared with the length of the crack: deep surface lakes storing much larger water volumes are beyond the scope of our work.*

8. Figure 4 and 5 are simply wonderful contributions to the literature on glacier fracture mechanics. Thank you for this. Thank you!

9. Figure 6 / Line 455. See comment above about the surface load due to a lake. As I understand it, the model essentially
140  has the water "coming from nowhere". Maybe the surface loading could resolve the paradox of stability at high prestress. There's an analytical SIF in Tada (2000) that you could compare to, see their section 8.9.

*As we point out immediately above, for the water volumes we have in mind here, with $O(1)$ values of $\beta$, changes in normal stress at the ice surface will be negligible. We suspect the comment is motivated by the effect of sizeable surface lakes on ice sheets. The volume contained in these seems to us likely to be much larger than a volume that could ever*
145  *by a crevasse that partially penetrates through the ice; effectively we would have $\beta$ much larger than any of the values shown in figure 6, and all but the shortest surface cracks necessarily unstable to full propagation through the entire ice thickness. Additional aspects of surface lake hydrofracture however really are not the purpose of the present paper, interesting though it would be to investigate that.*

10. Discussion, particularly the "problem" on Line 676: I think this same issue was discussed by Rist et al (2002). Their
150  solution was to introduce back stress from sidewall coupling. Maybe I'm wrong and they were solving a different problem, but either way I would appreciate a clarification.

Back stress is indeed likely to play a role in stabilizing ice shelves. However, near the calving front of an ice shelf, the very stress boundary condition that results from the usual imbalance between cryostatic and hydrostatic pressures at the calving front still dictates values of $\tau \approx 0.05$; back stress is a cumulative effect that reduces $\tau$ below the "unconfined
155  shelf" value of 0.05 as you move away from the calving front, but it does not help avoid the "problem" near the calving itself, since the above-critical values of $\tau$ there are simply the result of the boundary condition at the calving front.